# UNDERSTANDING THE EFFECTS OF RLHF ON LLM GENERALISATION AND DIVERSITY

**Robert Kirk**[*α] **Ishita Mediratta**[β] **Christoforos Nalmpantis**[β] **Jelena Luketina**[γ]

**Eric Hambro**[β] **Edward Grefenstette**[α] **Roberta Raileanu**[β]

[α] University College London, [β] Meta, [γ] University of Oxford

## ABSTRACT

Large language models (LLMs) fine-tuned with reinforcement learning from human feedback (RLHF) have been used in some of the most widely deployed AI models to date, such as OpenAI's ChatGPT or Anthropic's Claude. While there has been significant work developing these methods, our understanding of the benefits and downsides of each stage in RLHF is still limited. To fill this gap, we present an extensive analysis of how each stage of the process (i.e. supervised fine-tuning (SFT), reward modelling, and RLHF) affects two key properties: out-of-distribution (OOD) generalisation and output diversity. OOD generalisation is crucial given the wide range of real-world scenarios in which these models are being used, while output diversity refers to the model's ability to generate varied outputs and is important for a variety of use cases. We perform our analysis across two base models on both summarisation and instruction following tasks, the latter being highly relevant for current LLM use cases. We find that RLHF generalises better than SFT to new inputs, particularly as the distribution shift between train and test becomes larger. However, RLHF significantly reduces output diversity compared to SFT across a variety of measures, implying a tradeoff in current LLM fine-tuning methods between generalisation and diversity. Our results provide guidance on which fine-tuning method should be used depending on the application, and show that more research is needed to improve the tradeoff between generalisation and diversity.

## 1 INTRODUCTION

Large language models (LLMs) have become a standard approach to solving natural language processing (NLP) tasks. As these models become more capable, the tasks we want them to solve become more complex, which makes it more difficult to provide training data and to evaluate performance. For such tasks it may be easier and faster for humans to evaluate or rank model outputs than provide full demonstrations. Thus, there has been much recent work on using human preferences in this form to fine-tune LLMs, with one dominant approach being reinforcement learning from human feedback (Christiano et al., 2017; Ziegler et al., 2020, RLHF). This approach has been used to produce some of the most impressive AI systems that exist today (Glaese et al., 2022; OpenAI, 2022; 2023; Anthropic, 2023).

The standard RLHF fine-tuning pipeline generally consists of three stages: *supervised fine-tuning* (SFT), where the pretrained model is fine-tuned with the language modelling loss on demonstrations of the desired behaviour; *reward modelling* (RM), where the pretrained model is fine-tuned to predict human preferences between pairs of outputs for a given input; and *reinforcement learning* (RL), where the reward model is used to fine-tune the model produced by the SFT stage using an on-policy RL algorithm like PPO (Schulman et al., 2017). While this pipeline has been used to seemingly great success, there is little understanding about how each component contributes to the behaviour of the final model. Two important areas where the effects of each component of the pipeline have been underexplored are the out-of-distribution (OOD) generalisation and output diversity.

---

[*]Work partly done during an internship at Meta. Correspondence to `robert.kirk.20@ucl.ac.uk`. For details of author contributions, see Author Contributions.

OOD generalisation is important for the widespread adoption of these models, since it is necessary to ensure that LLMs are performant and reliable in a wide variety of situations that go beyond the distribution of the training data. While anecdotally models seem to perform well across a wide range of settings, it is unknown which stage of the pipeline is responsible for this strong generalisation, and even whether the observed generality is due to the training or fine-tuning methods used, the large model size, or purely from the very large and diverse distribution of data.

Further, these models are used in creative or open-ended domains such as story generation (Castricato et al., 2022), scientific research (Boiko et al., 2023), or other tasks where a diverse output distribution is required, such as red teaming (Perez et al., 2022). In these situations training models that produce diverse (but still high-quality) outputs is of crucial importance. There has been some speculation as to the possible effects of different steps of the RLHF pipeline on diversity (janus, 2022), and some work has shown a decrease in diversity from RLHF (Khalifa et al., 2021; Perez et al., 2022). However, no rigorous analysis has been done of the effects of different parts of the RLHF pipeline on output diversity across different tasks. In addition, in contrast with our paper, prior work has been limited to evaluating diversity using simple token-level metrics such as BLEU (Papineni et al., 2002) and on use cases which are not as common in practice.

In this work, we evaluate each stage of the RLHF pipeline (i.e. supervised fine-tuning, reward modeling, and reinforcement learning) as well as best-of-N (BoN) sampling in terms of their effects on in-distribution (ID) performance, OOD performance, and output diversity. We disentangle the effects of the data set and the training method on generalisation by using OOD test datasets that induce realistic distribution shifts between training and testing, and evaluate generalisation using these test sets at each stage of the pipeline (Section 5.1). While diversity is a difficult concept to operationalise, we take a pragmatic approach and measure a range of diversity metrics at each step of the pipeline, covering syntactic, semantic, and logical diversity (Section 5.2). We evaluate both diversity of outputs sampled for a single input and for a range of inputs. Evaluating BoN as well as SFT and RLHF enables us to uncover whether differences between RLHF and SFT are due to the use of a reward model or the type of optimisation applied.

In summary, we find:

- RLHF improves in-distribution performance (as expected from previous work) and also OOD performance in comparison to SFT.

- However, RLHF substantially decreases the diversity of outputs sampled for a given input compared to SFT.

- Even when sampling outputs for different inputs, RLHF produces less diverse text on some metrics, implying that such models tend to produce more similar text regardless of the input

These findings reveal an inherent tension between generalisation and diversity when applying current fine-tuning techniques. This underscores the necessity for novel methods that improve both these attributes without sacrificing one for the other, and for research to understand whether this tension is fundamental to fine-tuning or a deficit of current techniques. We open source our code to enable reproducible research here: `https://github.com/facebookresearch/rlfh-gen-div`.

## 2 BACKGROUND AND RELATED WORK

**Fine-tuning Large Language Models.** The current common practice in NLP is to fine-tune large pre-trained language models (LLM) for downstream tasks. The standard approach for fine-tuning is supervised fine-tuning (SFT), which trains the model on demonstrations of solving the task using supervised learning. When it is easier to evaluate or rank model outputs than it is to gather demonstrations that accurately perform the desired task, an alternative method called reinforcement learning from human feedback (Christiano et al., 2017; Ziegler et al., 2020, RLHF) can be used. Most previous work on RLHF uses on-policy RL algorithms such as Proximal Policy Optimization (PPO) (Schulman et al., 2017) or Advantage Actor Critic (A2C) (Mnih et al., 2016), but offline RL methods have also been proposed (Snell et al., 2022). Once a RM has been trained, it can also be used to do Best-of-N (BoN) sampling (also called rejection sampling) of the model outputs. (Casper et al., 2023) survey existing problems with RLHF as a method for fine-tuning LLMs, and our work further

investigates some of these problems (specifically policy generalisation and output diversity or "mode collapse").

Dubois et al. (2023, AlpacaFarm) introduces a framework for developing methods for learning from human feedback in an instruction following setting, and in this framework demonstrate that approaches that learn from human feedback (including RLHF) generally perform better than SFT on a specific evaluation set they introduce. We use the AlpacaFarm models and evaluation dataset in our experiments on instruction following. In that work they do not evaluate out-of-distribution generalisation or output diversity, and only present results on a single evaluation dataset, while we address all of these issues, while also performing the analysis in an additional task (summarisation).

We discuss other recent approaches for fine-tuning LLMs in Appendix C, but note that while these works sometimes show improvements, they are not used by most large-scale systems being deployed currently, and hence we focus our analysis on the more popular and widely used RLHF pipeline, as that is where understanding will be most relevant and useful.

**Generalisation and Diversity in NLP.** Almost all prior work using RLHF has evaluated models on the same distribution of inputs used for fine-tuning (Bai et al., 2022; Glaese et al., 2022; Ouyang et al., 2022; Stiennon et al., 2022), meaning that the generalisation properties of such methods isn't understood. One notable exception is Stiennon et al. (2022), who perform some experiments evaluating their models trained the TL;DR dataset (Völske et al., 2017) (reddit post summarisation) on the CNN/Daily Mail dataset (Nallapati et al., 2016) (news article summarisation). However, they didn't investigate how different parts of the pipeline affected generalisation, and the investigation was less rigorous and involved than ours is. (Hupkes et al., 2023) provide a comprehensive survey of generalisation in the wider NLP literature.

Several works (Khalifa et al., 2021; Perez et al., 2022) have shown in specific settings that RLHF fine-tuning produces models with less output diversity, as measured by self-BLEU (Zhu et al., 2018). Our work extends these works by making diversity evaluation a primary focus and using diversity metrics beyond self-BLEU which have been externally validated (Tevet & Berant, 2021) and measure diversity in a range of different ways.

We discuss more related work, including details on LLMs, SFT and RLHF, in Appendix C.

## 3  MODEL TRAINING

Here we briefly describe the details of how the models we evaluate are trained. For a more detailed description of model training see Appendix E.

**Pretrained Models.** We use the LLaMa pretrained 7 billion parameter model (Touvron et al., 2023a). This is a standard decoder-only transformer-based causal language model trained on a large corpus of web text. This size of model has been shown to be effective in the tasks we investigate (Stiennon et al., 2022; Dubois et al., 2023).

In Appendix J we perform experiments with OPT (Zhang et al., 2022) models, using five model sizes. These models have worse performance in general, but the experiments allow us to see trends across several model scales.

We train **Reward Models**(RMs) following (Stiennon et al., 2022). We initialise the RM as the base model, and we add a scalar head before the unembedding layer. The model is then fine-tuned on inputs and pairs of outputs with one output labelled as preferred, and trained to output a scalar for each input-output pair representing which output is preferred.

**Supervised Fine-Tuning**  (SFT) models are trained on reference input-output pairs using the cross-entropy loss on the output conditioned on the input.

**Reinforcement Learning from Human Feedback**  (RLHF) models are trained using PPO (Schulman et al., 2017) as our base RL algorithm, following previous work (Stiennon et al., 2022). We initialise the model with the corresponding SFT model, use the language modelling head as the policy output, and learn a value function using a linear layer (an identical architecture to the RM), with the policy and value function sharing the same model backbone. We optimise the policy to maximise the RM described above, and use a KL divergence term as an auxiliary reward to ensure that the language model stays close to the SFT model, as in previous work (Jaques et al., 2017; Stiennon et al., 2022;

Ziegler et al., 2020). The final reward for the policy is

$$R(x, y) = RM_{\theta_{RM}}(x, y) - \beta_{KL} D_{KL}(\pi_{\theta_{RL}}(y|x)||\pi_{\theta_{SFT}}(y|x)) \tag{1}$$

where $RM$ denotes the reward model trained as described above; $\theta_{RL}$, $\theta_{RM}$ and $\theta_{SFT}$ are the parameters of the policy, RM and SFT model respectively; $x, y$ are the input and output; and $\beta_{KL}$ is a hyperparameter that controls the weight of the KL penalty. We use $\beta_{KL} = 0.05$ throughout this work, following (Stiennon et al., 2022) and our own early experiments that found this choice struck a good balance between model performance and overoptimisation (Gao et al., 2022).

**Best-of-N.** The reward model can also be used to filter samples from another model; this is called Best-of-N (BoN) sampling and has been used in multiple previous works to achieve good performance (Menick et al., 2022; Nakano et al., 2022). $N$ summaries are sampled from the SFT model, and then the RM is used to select the best one. We sample with temperature 0.7 and use $N = 16$, as that is what is used in previous works (Rafailov et al., 2023) and strikes a good balance between improved performance and computational cost. Evaluating BoN performance gives use a way to evaluate whether the differences between RLHF and SFT models are due to the use of a RM, or the type of optimisation applied. However, due to the increased inference time compute cost, this method is generally not used in practice, so we focus our analysis on RLHF and SFT policies.

## 4 DATASETS AND TASKS

We investigate the effects of RLHF in two tasks: text summarisation and instruction following.

**Summarisation.** We follow Stiennon et al. (2022); Ziegler et al. (2020) in evaluating LLM fine-tuning algorithms on a summarisation task. Models are trained to produce summaries of Reddit posts, given the post as input. We use the same dataset as Stiennon et al. (2022), which is a filtered version of the TL;DR dataset (Völske et al., 2017), consisting of approximately 120,000 Reddit posts with accompanying summaries. See Stiennon et al. (2022) for full details on the filtering procedure and the analysis of the resulting data distribution.

We use the preference data gathered by Stiennon et al. (2022) for reward model training. This consists of approximately 64,000 summary comparisons. These data consist of inputs (reddit posts) along with pairs of outputs (summaries), with one summary labelled as preferred. The preferences are from human annotators contracted by Stiennon et al. (2022) to choose summaries according to a list of criteria designed to select high quality summaries.

**Instruction Following** (Chung et al., 2022; Dubois et al., 2023; Wang et al., 2022) is one of the main use cases for the LLM fine-tuning techniques we investigate, so we also evaluate models trained in this setting.

We use the SFT, RLHF, and RM models released by Dubois et al. (2023, AlpacaFarm). These models were trained in a very similar way to how we train our summarisation models, and all based on the LLaMa 7B base model. Models take a text instruction as input and are trained to output preferred answers to those instructions. The outputs used for the SFT model are from the `text-davinci-003` model from the OpenAI API, and the human preferences used to train the reward model are gathered by the authors of AlpacaFarm based on outputs of the SFT model. For precise details on how these models were trained, refer to Dubois et al. (2023).

## 5 MODEL EVALUATION

We now describe how we evaluate both out-of-distribution generalisation and output diversity for the different fine-tuning techniques.

### 5.1 GENERALISATION EVALUATION

To evaluate the performance of our trained models, we use GPT-4 (OpenAI, 2023) through the OpenAI API [1] as a *simulated human annotator*. While we ultimately care about human preferences, these are expensive, difficult to gather, and often noisy. Previous work has shown that GPT-4 accurately

---

[1]https://platform.openai.com/docs/guides/gpt

simulates human preferences in both summarisation (Rafailov et al., 2023) and instruction following (Dubois et al., 2023; Zheng et al., 2023), so we use it as the main performance metric in both tasks.

To evaluate a model's performance on a given dataset of inputs and reference outputs with GPT-4, we prompt GPT-4 with the input, the reference output, and the model output, and prompt it to decide which output is better. We use a variant of the prompts from (Rafailov et al., 2023) for summarisation, and alpaca_eval (Li et al., 2023) with the standard prompts for instruction-following. See Appendix D for the precise prompts and other details. This gives us a *percentage win rate of the model being evaluated versus the human-annotated reference output*, which we refer to as preference vs reference (**PvR**). In Appendix D.1 we validate that this evaluation is a good proxy for human preferences. We also perform *head-to-head comparisons between two policies*, by prompting GPT-4 in the same way to decide which of two model model outputs is better.

To evaluate out-of-distribution (OOD) generalisation, we specify an in-distribution (ID) test set and one or more out-of-distribution (OOD) test sets for each task, which have inputs drawn from a different distribution. In each of these sets we have evaluation inputs and corresponding reference outputs (produced either by humans in summarisation or `text-davinci-003` in instruction-following).

**For summarisation**, *the ID test set is the original TL;DR* test set from (Stiennon et al., 2022), and *the OOD test set is the CNN/DailyMail* test set, a dataset of news articles and corresponding summaries (Nallapati et al., 2016). This tests the ability of the model to have learnt the more general skill of summarisation and to apply it in a very different domain.

**For instruction following**, the *ID test set is a new test set generated in the same way as the training set was for AlpacaFarm, using the AlpacaFarm variant of Self-Instruct* (Dubois et al., 2023; Wang et al., 2023). Regenerating the test set ensures that it was not seen during training or model selection for AlpacaFarm models. For *the first OOD test sets, we use the AlpacaEval evaluation test set proposed in the original paper*. This is a set of inputs taken from a variety of open-source instruction following and dialogue training and evaluation datasets (Bai et al., 2022; Ganguli et al., 2022; Gudibande, 2023; Köpf et al., 2023; Wang et al., 2023; Zheng et al., 2023), curated by Dubois et al. (2023). For an *additional OOD test set, we generate a set of Sequential Instructions using an adjusted Self-Instruct protocol*. These instructions contain multiple steps in a single input, often building on each other, and require the model to ensure that they complete all the steps to produce a satisfactory outcome. For more details on this dataset see Appendix H.

We also report the *generalisation gap* (the difference between in-distribution and out-of-distribution performance) which provides a measure of generalisation specifically. Lower generalisation gaps imply the model generalises better, as model performance does not drop as much when the model is evaluated out of distribution. For the head-to-head comparisons between models, we look at the change in head-to-head winrate to give different evaluation of generalisation; the model whose winrate increases generalises better.

## 5.2 DIVERSITY EVALUATION

To evaluate the output diversity of trained policies, we use several diversity measures which are well-supported by prior work, namely **distinct N-grams** (Li et al., 2016), **Sentence-BERT embedding cosine similarity** (Reimers & Gurevych, 2019) and **NLI diversity** (Stasaski & Hearst, 2022). All of these metrics have been shown to align well with human diversity evaluations and with underlying diversity generators by Tevet & Berant (2021).

Each diversity metric $D$ takes a set of model outputs $O$, and produces a scalar score representing how diverse the set is. *Distinct N-grams* counts the number of distinct N-grams (averaging over $n = 1 \ldots_5$) in the set of outputs, and following (Liu et al., 2022) we use the expectation-adjusted distinct N-grams (EAD) formula to remove the bias towards shorter outputs. The *Sentence-BERT* metric embeds each element of the output set using a sentence transformer, and then measures the average cosine similarity between the embeddings. The metric is then 1 minus the average similarity.

The *NLI diversity* metric measures the number of entailments and contradictions between pairs of elements in the output set using a natural language inference (NLI) model and rates an output set as more diverse if it produces more contradictions and fewer entailments. We pass sentences sampled from elements of the output set (rather than complete elements) to the NLI model so that the inputs are closer to the model's training distribution.

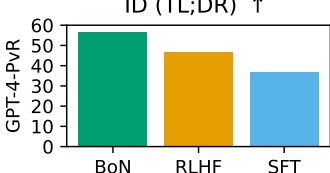 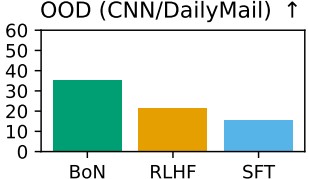 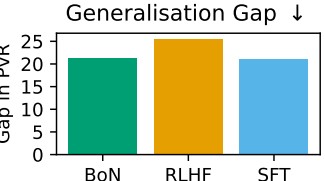

Figure 1: **Summarisation Generalisation Results.** GPT-4-PvR for SFT, BoN and RL policies, based on LLaMa 7B, trained on the summarisation task. In-distribution is performance on TL;DR, and out-of-distribution is on CNN/DailyMail, and generalisation gap is ID – OOD performance.

For each policy $\pi$ we produce a set of $K$ outputs from the model for each of a subset of $N$ inputs from the test set, sampling with temperature 1:

$$\text{for } i = 1 \ldots N : \mathcal{O}_\pi^i := \{y_j \sim \pi(y|x_i)|j = 1 \ldots K\}.$$

In this work we use $K = 16$, $N = 500$. For a diversity metric $D$ we then evaluate the **per-input diversity** which is defined as the average diversity of the output sets over inputs (i.e. the diversity of $\pi(y|x)$), as well as **across-input diversity** defined as the diversity of outputs across inputs (i.e. the diversity of $\pi(y)$):

$$\text{PerInputDiversity}_D(\pi) := \frac{1}{N} \sum_{i=1}^{N} D(\mathcal{O}_\pi^i) \quad \text{CrossInputDiversity}_D(\pi) := D\left(\bigcup_{i=1}^{N} \mathcal{O}_\pi^i[1]\right)$$

$$(2) \qquad\qquad\qquad\qquad\qquad\qquad\qquad (3)$$

Here $\mathcal{O}_\pi^i[1]$ is the first element of the set $\mathcal{O}_\pi^i$. For conciseness, we refer to expectation-adjusted distinct N-grams, sentence-BERT average cosine similarity and NLI diversity as EAD, Sent BERT and NLI respectively. We can view them as measuring *syntactic, semantic and logical diversity*, and hence using all of them ensures that we are evaluating diversity in a wide range of ways.

## 6 EXPERIMENTAL RESULTS

In this section we present the results of our experiments on generalisation and output diversity. In general we find that RLHF performs better than SFT in an absolute sense both ID and OOD, but generalisation gap and head-to-head metrics tell a more nuanced story. However, RLHF does reduce output diversity substantially in the per-input setting, and still reduces it to a lesser extent in the cross-input setting. These conclusions are supported by similar results for OPT models across a range of model scales in the summarisation task, presented in Appendix J.

### 6.1 GENERALISATION

**Summarisation.** Fig. 1 shows the *GPT4-PvR* for SFT, Bo16 and RLHF policies trained on the summarisation task, showing the ID (TL;DR) and OOD (CNN/DailyMail) performance and the generalisation gap; Bo16 outperforms RLHF which outperforms SFT, both ID and OOD. While Bo16 outperforming RLHF is somewhat surprising, there are examples of this in the literature (Nakano et al., 2022), and Bo16 has a substantially higher inference-time cost, making RLHF better for practical applications. The performance of all policies drops OOD, which is unsurprising given the difficulty of the shift (reddit post summarisation to news article summarisation). The generalisation gap is fairly similar between methods, implying that none of these methods has a particular advantage with respect to generalisation specifically in this setting.

These results also show that the generalisation gap for SFT and Bo16 policies are the same (and they are still the same for a different choice of temperature for Bo16, see Fig. 8). This can be explained by the fact that the reward model generalises near-perfectly to the CNNDM preference dataset from (Stiennon et al., 2022). ID and OOD accuracy is 75.8 and 71.6 respectively, and considering that the maximum inter-annotator agreement in the CNNDM preference dataset is lower than TL;DR (and hence the maximum accuracy attainable is lower), it is plausible that the RM is not suffering any real

drop in OOD performance in this case. This implies that all of the drop in performance for Bo16 is driven by the drop in performance of the SFT model, as if both SFT and RM performed worse OOD we would expect those drops in performance to compound. Overall, this shows that if you can expect your reward model to generalise well then BoN is a good choice of policy, although it is limited by the generalisation abilities of the underlying model being sampled from (the SFT model in this case), and is more expensive at inference time than RLHF and SFT.

In Appendix J we present results for a range of models based on OPT (Zhang et al., 2022), trained in a similar way to the LLaMa models but on split versions of the TL;DR dataset, and evaluated with a LLaMa proxy reward model. These results show similar trends: BoN outperforms RLHF which in turn outperforms SFT at the largest model size, and the ordering holds OOD for 3 different splits of the training dataset. This shows our results are robust across different evaluation metrics and base models.

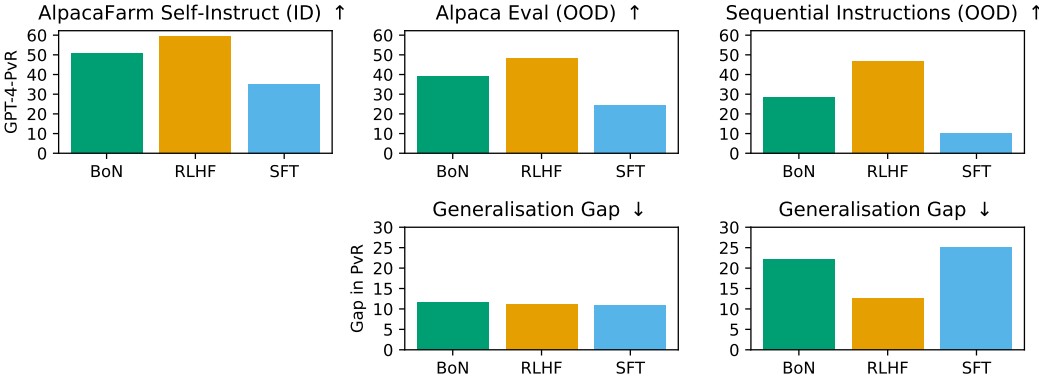

Figure 2: **Instruction Following Generalisation Results.** GPT-4 PvR for SFT, BoN and RL policies, based on LLaMa 7B, trained on the AlpacaFarm Self-Instruct instruction following task. ID is on AlpacaFarm Self-Instruct, OOD is on the AlpacaEval and Sequential Instructions datasets respectively, and generalisation gap is ID – OOD performance.

**Instruction Following.** Fig. 2 shows the results of BoN, SFT and RLHF models trained in AlpacaFarm Self-Instruct (Dubois et al., 2023) evaluated ID and OOD on AlpacaEval and Sequential Instructions. Similar to summarisation, we see that RLHF and Bo16 both outperform SFT, but here RLHF outperforms Bo16 across all datasets, in contrast to the summarisation task. As the focus of this paper is mostly comparing RLHF and SFT, we did not investigate this result further, but there could be many possible reasons for the change in ordering between RLHF and Bo16, as the two tasks are very different and the model training procedures are not identical.

We see that on AlpacaEval (the easier OOD generalisation task), models all generalise equally well, but on the harder Sequential Instructions OOD task, RLHF generalises much better. This suggests that *RLHF may generalise better relative to SFT for larger distribution shifts*, which potentially explains why models fine-tuned with RLHF have been observed to be much better in practice when interacting with users (Touvron et al., 2023b; Ouyang et al., 2022, *inter alia*): when users interact with these models the distribution shift is quite pronounced and hence many inputs are more OOD, and this is where RLHF model performance continues to be high.

While GPT-4 PvR is a useful metric, it does not show a difference in generalisation (as measured by generalisation gap) between SFT, RLHF and Bo16 models on the easier AlpacaEval dataset. This could be due to these models having similar generalisation properties, or be a deficiency of the metric. To investigate this further, we use can look at the GPT-4 Head-to-Head winrate of SFT vs Bo16 and vs RLHF, which is shown in Fig. 3. These results show that both RLHF and Bo16 winrates *improves* vs SFT by approx-

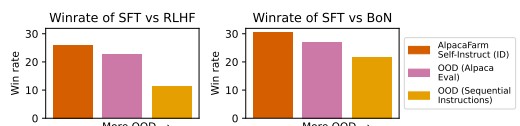

Figure 3: **GPT-4 Head-to-Head Winrate** of SFT vs RLHF and Bo16 in AlpacaFarm Self-Instruct, AlpacaEval and Sequential Instructions datasets.

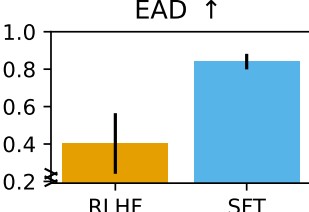 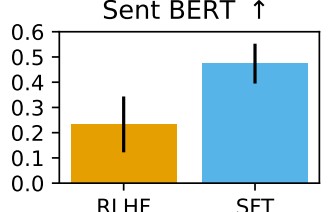 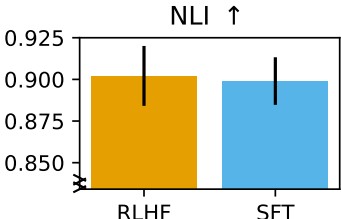

Figure 4: **Per-input diversity metrics for RLHF and SFT models**. For these scores the outputs used to calculate the diversity are a sample of outputs from the model for single input. These per-input scores are then averaged, as in Eq. (2). Error bars are standard deviation of the per-input diversity score across different inputs. Note that some plots have broken y-axis for better visualisation.

imately 3.5% from ID to AlpacaEval OOD. This implies that RLHF and Bo16 generalise better than SFT even in this case, emphasising the need for a range of metrics when evaluating model generalisation.

## 6.2 DIVERSITY

For the diversity evaluations, we focus on the summarisation task specifically, as it has the most compelling results. We ran some initial experiments evaluating diversity for the instruction-following models, but we did not see any meaningful differences. We hypothesise this is due to the diversity metrics we use being designed for settings where the model output is relatively short (e.g. a single sentence), whereas in the instruction-following setting outputs are generally much longer. Furthermore, RLHF models tend to produce longer outputs than other models, which can confound the evaluation of output diversity, since most metrics are not invariant to output length.

Fig. 4 shows the per-input diversity scores for RLHF and SFT models in the summarisation task. We see that across the first two metrics, RLHF has much lower output diversity than SFT. Fig. 5 shows the across-input diversity scores in the same setting. Here we see that while SFT generally has slightly higher diversity as before, the difference is much smaller than in the per-input case. The drop in across-input diversity cannot be explained purely by the use of the reward model, as BoN has similar or higher across-input diversity that SFT for the first two metrics. Both these trends are the same for OPT across model sizes (see Appendix J.4 for full results) We find that NLI does not show meaningful difference between models in either per-input or across input, showing that in a logical sense all models are similarly diverse.

The difference in across-input diversity between RLHF and SFT, while small, can also be taken as evidence of the phenomena of "mode collapse" hypothesised to occur under RLHF fine-tuning (janus, 2022). The hypothesised effect is that even for different inputs, RLHF models can be biased towards outputting text of a specific style or "mode", meaning that even changing the inputs to a model is not sufficient to generate truly diverse outputs. We believe that this is the first rigorous empirical demonstration of across-input mode collapse emerging from RLHF training specifically.

For most metrics and models, the across-input diversity scores are higher than the per-input diversity scores, which is expected given the across-input diversity distribution is much broader. However, for EAD (which measures diversity at the n-gram level), SFT has similar levels of per-input and across-input diversity. This is likely due to SFT effectively approaching the maximum EAD diversity even in the per-input case, so that the across-input diversity cannot be much higher.

## 6.3 THE IMPACT OF THE KL PENALTY

We have shown that while RLHF improves performance ID and OOD in an absolute sense, this comes at the cost of substantial drops in output diversity relative to SFT. Motivated by the fact that the KL penalty coefficient (see Eq. (1)) encourages the RLHF policy to stay closer to the SFT policy, we investigate whether adjusting this coefficient trades off between generalisation and diversity. The results show that this does not work – *increasing the KL penalty coefficient leads to a drop in performance as expected, but also to a drop in per-input diversity*, rather than a gain (see Appendix I

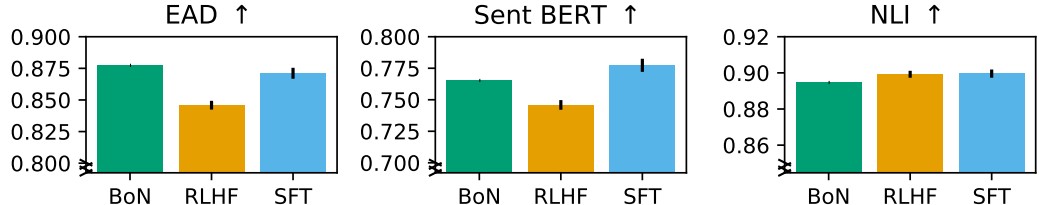

Figure 5: **Across-input diversity metrics for RLHF, BoN and SFT models**. For these scores the outputs used to calculate the diversity are a set of single outputs from a range of inputs, as in Eq. (3). Note that all plots have broken y-axis for better visualisation; the differences between SFT and RLHF are much smaller in this case than in the per-input diversity metrics in Fig. 4. Error bars (where present) are standard deviation of the across-input scores over different samples from the set of outputs for each input.

for details). This emphasises that more research is needed to investigate whether more sophisticated methods can improve the trade-off between generalisation and diversity.

# 7 DISCUSSION AND CONCLUSION

**Summary of Contributions.** In this work, we analyse three methods for fine-tuning LLMs (RLHF, SFT, and BoN) across two problem settings (summarisation and instruction following) in terms of OOD generalisation and output diversity. We demonstrate an inherent tradeoff between generalisation performance and output diversity when choosing between these methods: RLHF produces more performant models both in-distribution and out-of-distribution, but at the cost of lower output diversity, both per-input and across-input. It is unclear whether this tradeoff is a fundamental one in fine-tuning LLMs with RLHF or just demonstrates a deficiency in current methods. We suspect the answer will be a combination of both explanations: There will be a pareto-frontier of output diversity vs generalisation performance on which tradeoffs have to be made, but current methods do not yet seem to be at that frontier. Future work could investigate producing methods that are closer to this frontier, either through increasing the performance of SFT or increasing the output diversity of RLHF.

When looking at generalisation metrics that control for ID performance, results are mixed. RLHF generalises better for the most difficult distribution shift in the instruction following setting, but in less difficult shifts RLHF generalises similarly or slightly worse than SFT (as measured by generalisation gap and head-to-head performance drop). While RLHF still performs best OOD in absolute terms, these results demonstrate the need for the multifaceted evaluation we perform in this paper as opposed to focusing on a single metric of performance.

**Implications for Practical Applications.** Our results have implications for which fine-tuning method should be used in different situations. The OOD performance of RLHF on the most difficult instruction following task is evidence for the utility of RLHF when large distribution shifts are likely to occur, such as when training models to be used as chat bots by users (OpenAI, 2022; Anthropic, 2023). However, in use cases where the model needs to generate a wide variety of outputs, such as story generation (Castricato et al., 2022), red-teaming (Perez et al., 2022), and when doing rejection sampling (Cobbe et al., 2021), supervised fine-tuning may be desirable. In cases where you can expect the reward model to generalise very well (for example, it is likely easier to spot whether text is toxic or not than to never generate toxic text), then BoN may produce better generalisation results, although its performance will always be limited by the generalisation of the underlying model being sampled from, and its inference time cost is much greater than that of SFT or RLHF models.

**Future Research Directions.** This work also suggests areas for further research. Future work should investigate *why* RLHF reduces the output diversity so much, and whether this diversity can be recovered without the loss of performance. Inspiration could be taken from the deep reinforcement learning literature, where several works specifically inject diversity into the RL optimisation process to increase the policy's diversity Eysenbach et al. (2019); Haarnoja (2018); Osa et al. (2022); Kumar et al. (2020). Also, while there are some hypotheses about why RLHF generalises better than SFT (Goldberg, 2023), it is important to experimentally validate these in order to build our understanding of how these methods work and when they should be used.

## ACKNOWLEDGEMENTS

We would like to thank (in alphabetical order) Akbir Khan, Amy Zhang, Dieuwke Hupkes, Ethan Perez, Jacob Hilton, Kyle McDonnell, Laura Ruis, Louis Castricato, Patrick Lewis, Sebastian Riedel, Stephen Roller, Susan Zhang, Tim Rocktäschel and Verna Dankers for discussions and feedback on ideas related to this project. Robert Kirk is supported by the UCL CDT in Foundational AI.

## AUTHOR CONTRIBUTIONS

**Robert Kirk** lead the project, set the direction, designed, programmed and ran the majority of the experiments, and wrote much of the paper. **Ishita Mediratta** assisted with programming and running the RLHF training experiments. **Christoforos Nalmpantis** programmed parts of the GPT-4 evaluation code and initial RLHF training code. **Jelena Luketina** assisted in the AlpacaFarm evaluations. **Eric Hambro** programmed earlier versions of the RLHF training code. **Edward Grefenstette** advised on project direction and paper writing. **Roberta Raileanu** advised on project direction, experiment design, programming and paper writing. All authors participated in discussions over experiment design and paper editing.

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

## A LIMITATIONS AND FUTURE WORK

Here we discuss some potential limitations of our work and possible future directions for research pointed to by our results. While our work shows the effects of RLHF on generalisation and output diversity empirically, we do not provide a theoretical explanation for these results. Furthermore, while we demonstrate results on multiple base models and tasks, more combinations of base models and tasks could be experimented on, as well as other methods. Future work could investigate whether these effects are more general and why they arise.

Our work also only investigates SFT, RLHF and BoN as methods for fine-tuning language models with human preferences, but there are many other methods as described in Appendix C. Understanding the effects of these methods on generalisation and output diversity would be beneficial, especially if some of those methods are able to provide the generalisation benefits of RLHF without harming output diversity to the same extent.

Finally, we only evaluate our models on automatic metrics and do not perform any human evaluation (although we validate that our metrics align well with human preferences in Appendix D.1). While automatic metrics are useful for comparing models, they are not a perfect proxy for human judgement. Future work could investigate the effects of RLHF on human judgement of model outputs.

## B BROADER IMPACT

Large Language Models are increasingly used in production systems, and so it is important to understand the effects of different fine-tuning methods on the properties of the resulting models. Our work shows that RLHF produces better-generalising models than SFT, but those models are less diverse. This could be beneficial for some use cases, but harmful for others. For example, if a model is being used to generate text for a chatbot, it is important that the model is able to generalise to new inputs, but also that it is able to produce diverse outputs. On the other hand, if the model is being used to generate text for a summarisation system, it is important that the model is able to generalise to new inputs, but less important that it is able to produce diverse outputs.

## C RELATED WORK

**More RLHF and SFT details**  Often, SFT and RLHF are combined by performing SFT followed by RLHF (Glaese et al., 2022; Menick et al., 2022; Nakano et al., 2022; Ouyang et al., 2022). We call the process of doing SFT followed by RLHF "The RLHF Pipeline", as it's the standard approach used in the literature and in deployed products (that use RLHF) (OpenAI, 2022; Anthropic, 2023). Other work has directly used RLHF on top of a prompt-distilled language model (Bai et al., 2022). Prompt distillation gathers demonstrations from a prompted version of a base model, and then performs SFT on the base model with these outputs, effectively fine-tuning the model to behave as if it was always prompted.

Ramamurthy et al. (2022) introduced an adaptation of PPO specifically for RLHF called *Natural Language Policy Optimisation* (NLPO), which calculates an action mask with top-p sampling and applies this to the language model, resulting in slightly improved performance on a range of tasks when using automated reward functions (not trained with human preferences for the task). Ramamurthy et al. (2022) demonstrate that RL generally outperforms SFT, but that their combination performs the best. However, their work only investigates relatively small models (220 million parameters), does not evaluate OOD performance or diversity, and does not use reward functions trained with human feedback, instead using mostly hard-coded reward functions from the literature. While hard-coded reward functions can sometimes be useful, RLHF is most widely applied in settings where we do not have a hard-coded reward function, and hence need to learn one from human data.

**Large Language Models**  (LLMs) have recently been shown to solve a wide variety of language-based tasks, often without additional gradient-based training. Examples of such models include GPT-3 (Brown et al., 2020), Gopher(Rae et al., 2022), Chinchilla (Hoffmann et al., 2022), OPT(Zhang et al., 2022) PaLM (Chowdhery et al., 2022), Claude (Anthropic, 2023) and LLaMa(Touvron et al., 2023a). These models are trained on large amounts of text data, with a simple language modelling objective, and can often be prompted to perform tasks zero-shot or few-shot, without additional

training (Brown et al., 2020). Such tasks include translation, question-answering, and other standard NLP tasks, as well as newer tasks such as using LLMs to simulate human annotators (Dubois et al., 2023; Mao et al., 2023; Liu et al., 2023b) or as content generators for improving other models (Peng et al., 2023; Wang et al., 2023).

**Other methods for fine-tuning LLMs** Recently, multiple approaches for fine-tuning LLMs have been proposed: *Chain of Hindsight* (Liu et al., 2023a) fine-tunes models using SFT on sequences of increasingly better outputs for a given input; *CoOp CARP* (Castricato et al., 2022) uses a dataset of story-critique pairs combined with contrastive learning and pseudo-labelling to learn a preference model that is then used in the RLHF pipeline; *RRHF* (Yuan et al., 2023), uses a RM to rank multiple outputs from the model, and then optimises the model with weighted SFT, with a negative weight (similar to unlikelihood training (Welleck et al., 2020)) on lower-ranked samples; *HIR* (Zhang et al., 2023) relabels outputs using a goal-conditioned reward function or feedback function and then trains a goal-conditioned policy on these outputs (similar to (Andrychowicz et al., 2017)); and *ILF* (Scheurer et al., 2023), which uses natural language human feedback to prompt the model to produce better outputs than its original inputs, and then optimises the model with SFT on this dataset of improved outputs. While these works sometimes show improvements, they are not used by most large-scale systems being deployed currently, and hence we focus our analysis on the more popular and widely used RLHF pipeline, as that is where understanding will be most relevant and useful.

**Possible Explanations for Results.** Xu et al. (2022) present results for adversarial imitation learning (AIL) as compared to behavioural cloning (BC) in a more classical RL setting, showing that often AIL methods can generalise better than BC because they optimise the policy on out-of-distribution states. Mapped to the LLM fine-tuning regime, AIL is somewhat analogous to RLHF, and BC is identical to SFT, so this result may somewhat explain why RLHF generalises better than SFT.

Goldberg (2023) hypothesises that RLHF may generalise better than SFT because RLHF does not force models to produce outputs that are not implied their internal world model (to the extent that exists), whereas SFT trains models to produce outputs even if the model "believes" those outputs to be false.

## D  GPT-4 EVALUATION DETAILS

For GPT-4 evaluations for both summarisation and instruction following, we use the AlpacaEval (Li et al., 2023) software package to query GPT-4. For the instruction-following prompts, we use the standard annotator configuration recommended for that dataset, `alpaca_eval_gpt4`. For summarisation, we use the same configuration, but change the prompts to utilise a variant of those provided in (Rafailov et al., 2023). For the exact prompts see Figs. 6 and 7.

### D.1  VALIDATING GPT-4 EVALUATION

**Summarisation.** We validate the use of our GPT-4 evaluator for summarisation in two ways. First, we use the evaluator to label the preference validation datasets for both TL;DR and CNN/DailyMail released by Stiennon et al. (2022) and measure their accuracy. On TL;DR our evaluators gets 71.7% accuracy and on CNN/DailyMail it gets 65.5% accuracy. Comparing this to the inter-annotator agreement reported by Stiennon et al. (2022) or $\tilde{7}0\%$ for TL;DR and $\tilde{6}6\%$ for CNN/DailyMail demonstrates that our annotator has strong agreement with the human raters that generated the preference data we use for training our reward models.

For the second validation of GPT-4 as an evaluator, we measure the agreement between human labellers and GPT-4 on a subset of 25 inputs for every test set we use, comparing both SFT and RLHF model outputs with the reference output. This results in a total of 100 datapoints, each labelled by two human labellers giving 200 total annotations. This tests whether GPT-4 is in agreement with human preferences on the models we evaluate in this work.Table 1 shows the preference rating for GPT-4 and human labellers for each dataset and model, and the agreement between labellers and GPT-4, and we see that both at an aggregate level and at a per-example level our GPT-4 evaluator has good agreement with expert labellers.

```
<|im_start|>system
You are a helpful assistant, that ranks models by the quality of their
answers.
<|im_end|>
<|im_start|>user
Which of the following summaries does a better job of summarizing the most
important points in the given news article, without including unimportant
or irrelevant details? A good summary is both precise and concise.
Post: """{instruction}"""
Summary A:
{
    "model": "model_1",
    "summary": """{output_1}"""
}
Summary B:
{
    "model": "model_2",
    "summary": """{output_2}"""
}

Now please rank the models by the quality of their summaries, so that the
model with rank 1 has the best summary. Then return a list of the model
names and ranks, i.e., produce the following output:
[
    {'model': <model-name>, 'rank': <model-rank>},
    {'model': <model-name>, 'rank': <model-rank>}
]

Your response must be a valid Python dictionary and should contain nothing
else because we will directly execute it in Python. Please provide the
ranking that the majority of humans would give.
<|im_end|>
```

Figure 6: GPT-4 evaluation prompt for the CNN DailyMail dataset.

Table 1: GPT-4 agreement with human raters at an aggregate and individual level, across both summarisation datasets and RLHF and SFT model types. We see that both at an aggregate level and at a per-example level our GPT-4 evaluator has good agreement with expert labellers.

| Dataset | TL;DR | | CNNDM | |
| Model Type | RLHF | SFT | RLHF | SFT |
|---|---|---|---|---|
| GPT-4 winrate | 0.40 | 0.40 | 0.24 | 0.08 |
| Human winrate | 0.48 | 0.40 | 0.30 | 0.30 |
| H-GPT-4 agreement | 69% | 72% | 74% | 72% |

```
<|im_start|>system
You are a helpful assistant, that ranks models by the quality of their
answers.
<|im_end|>
<|im_start|>user
Which of the following summaries does a better job of summarizing the most
important points in the given forum post, without including unimportant
or irrelevant details? A good summary is both precise and concise.
Post: """{instruction}"""
Summary A:
{
    "model": "model_1",
    "summary": """{output_1}"""
}
Summary B:
{
    "model": "model_2",
    "summary": """{output_2}"""
}

Now please rank the models by the quality of their summaries, so that the
model with rank 1 has the best summary. Then return a list of the model
names and ranks, i.e., produce the following output:
[
    {'model': <model-name>, 'rank': <model-rank>},
    {'model': <model-name>, 'rank': <model-rank>}
]

Your response must be a valid Python dictionary and should contain nothing else
because we will directly execute it in Python. Please provide the ranking that
the majority of humans would give.
<|im_end|>
```

Figure 7: GPT-4 evaluation prompt for the TL;DR dataset.

**Instruction Following.** For instruction following, we use the evaluator released in (Li et al., 2023), which has been rigorously show to agree well with human labellers both at a per-example and aggregate level. Hence, we do not validate this ourselves.

# E MODEL TRAINING DETAILS

## E.1 REWARD MODEL TRAINING.

To train the Reward Model (RM), we again follow (Stiennon et al., 2022). We initialise the RM as the base model, and we add a scalar head before the unembedding layer. The model is then fine-tuned on inputs and pairs of outputs with one output labelled as preferred. The RM takes the full input and output, and outputs a scalar value. This gives us a scalar value for each output in the pair, and these values are treated as logits in a softmax to predict the correct preference label. We train the RM with a cross-entropy loss. Formulating the RM in this way means it can (in theory) learn to predict any transitive ranking over possible outputs, while still maintaining a type signature suitable for use as a reward function in an RL context (producing a scalar value given a single input-output pair which acts as a proxy for the quality of the output given the input, as evaluated by human annotators).

## E.2 POLICY TRAINING

We treat the autoregressive language model as a reinforcement learning policy, where the action space is the set of possible tokens, the state space is the current input, and the transition function adds the outputted action to the input. The episode terminates when the maximum number of tokens is generated, or the model outputs an end-of-sequence token.

**Supervised Fine-Tuning.** In this interpretation of the LLM as a policy, Supervised Fine-Tuning (SFT) can be seen as behavioural cloning, a form of imitation learning in RL, where the behaviour being cloned is that of the human who produced the original output. The policy is trained with the cross-entropy loss on batches of inputs concatenated with outputs, with the loss calculated only on the output tokens.

**Reinforcement Learning from Human Feedback.** Again following previous work we use PPO (Schulman et al., 2017) as our base RL algorithm. We initialise the model with the corresponding SFT model. We treat the language modelling head as the policy output, and learn a separate value function which takes the final hidden state of the language model and outputs a scalar using a MLP layer (an identical architecture to the reward model). We use a shared backbone for the policy and value function, with only the two heads being different. We train with the reward function described in Appendix E.1, and use a KL divergence term as an auxiliary reward to ensure the language model stays close to the SFT model, as in prior work (Jaques et al., 2017; Stiennon et al., 2022; Ziegler et al., 2020). The final reward for the policy is

$$R(x, y) = RM_{\theta_{RM}}(x, y) - \beta_{KL} D_{KL}(\pi_{\theta_{RL}}(y|x) || \pi_{\theta_{SFT}}(y|x)) \tag{4}$$

where $RM$ denotes the reward model trained as described in Appendix E.1; $\theta_{RL}$, $\theta_{RM}$ and $\theta_{SFT}$ are the parameters of the policy, reward model and SFT model respectively; $x, y$ are the input and output; and $\beta_{KL}$ is a hyperparameter that controls the weight of the KL penalty. We use $\beta_{KL} = 0.05$ throughout this work, following (Stiennon et al., 2022) and our own early experiments that found this choice struck a good balance between model performance and overoptimisation (Gao et al., 2022).

**Best-of-N.** Instead of using the RM to train a policy via PPO, the reward model can be used to filter samples from another model; this is called Best-of-N (BoN) sampling, and has been used in multiple previous works to achieve good performance (Menick et al., 2022; Nakano et al., 2022). $N$ summaries are sampled from the pretrained model, and then the RM is used to select the best one. This method removes the need to perform RL policy training, but makes inference time more computationally expensive, as we require $N$ generations from the policy and $N$ passes through the reward model to produce a single output. In this work we do BoN sampling on top of the SFT model, and mostly use $N = 16$, as that is what is used in previous works and strikes a good balance between improved performance and computational cost.

We summarise the core differences in training from (Stiennon et al., 2022) in Appendix F.

Table 2: Hyperparameters for SFT model training. These are fixed across all dataset splits and model sizes and types for summarisation.

| Hyperparameter | Value |
|---|---|
| batch size | 128 |
| epochs | 1 |
| adam beta1 | 0.9 |
| adam beta2 | 0.999 |
| adam epsilon | 1e-8 |
| frozen layers | 80% |

Table 3: Hyperparameters for RM training. These are fixed across all dataset splits and model sizes and types for summarisation.

| Hyperparameter | Value |
|---|---|
| batch size | 64 |
| epochs | 1 |
| adam beta1 | 0.9 |
| adam beta2 | 0.999 |
| adam epsilon | 1e-8 |
| frozen layers | 80% |

### E.3    MODEL SELECTION

For the results reported in the paper, we sweep over 3-5 learning rates for each model type, and select the best model on the validation set using an appropriate metric (accuracy for RMs, loss for SFT, reward for RL) - see Appendix E.4. For both the in-distribution and out-of-distribution results, we always use a validation set drawn from the same distribution as training. This more closely matches a real-world setting where we would not have access to OOD data to do model selection, and would have to draw model selection data from the same distribution used in training.

### E.4    HYPERPARAMETERS

**Summarisation.** For each model type (SFT, RM, RLHF) we do a sweep over learning rate, choosing ranges of values informed by choices in previous work (Stiennon et al., 2022) and early experimentation. The results in the paper are the best model with the learning rate chosen on an in-distribution validation set using loss, accuracy and reward respectively for SFT, RM and RLHF training. The learning rates for SFT are 3e-4, 1e-4, 3e-5, with 3e-5 selected; for RMs are 3e-4, 1e-4, 3e-5, 1e-5, 3e-6, with 3e-5 selected; for RLHF are: 1.5e-6, 3e-6, 6e-6, 1.5e-5, 3e-5, with 1.5e-5 selected.

We list the other hyperparameters (which are unchanged between all runs) for SFT, RM and RLHF training in Table 2, Table 3 and Table 4 respectively. We chose these following prior work (Stiennon et al., 2022).

**Instruction Following.** For the instruction following results, we use the models released by (Dubois et al., 2023), and so the hyperparameters can be found in that work.

## F    DIFFERENCES FROM STIENNON ET AL. (2022)

As we mostly follow (Stiennon et al., 2022) in the training of our summarisation models, we here describe the main differences between our work and theirs in terms of training. For RLHF, we train a single model with policy and value head, rather than separate policy and value functions. This is much more computationally efficient, and follows other recent work that still achieves impressive results (Glaese et al., 2022). This means that we randomly initialise the value function head, rather than intialising the value function from the reward model as is done in (Stiennon et al., 2022). We use LLaMa (Touvron et al., 2023a) (and OPT (Zhang et al., 2022) in Appendix J) as our pretrained

Table 4: Hyperparameters for RLHF model training. These are fixed across all dataset splits and model sizes and types for summarisation. One PPO step consists of generating *batch size* samples, and then performing *ppo epochs* of optimization on them, split into *ppo minibatch size* minibatches.

| Hyperparameter | Value |
|---|---|
| batch size | 256 |
| ppo epochs | 4 |
| ppo steps | 750 |
| ppo minibatch size | 256 |
| KL penalty coefficient | 0.05 |
| normalise advantages | True |
| adam beta1 | 0.9 |
| adam beta2 | 0.999 |
| adam epsilon | 1e-8 |
| frozen layers | 80% |

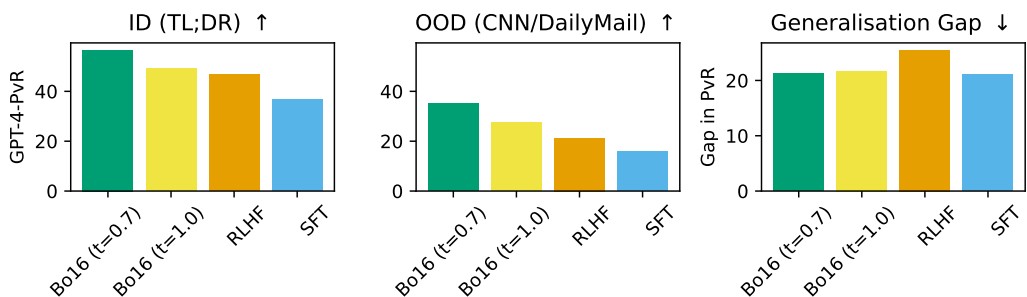

Figure 8: GPT-4 API evaluation win rate vs reference summaries for SFT, Bo16 with two different temperatures, and RL models, trained on the summarisation task. In-distribution is performance on TL;DR, and out-of-distribution is on CNN/DailyMail. Effectively a version of Fig. 1 with addition Bo16 results with a worse-performing temperature

models, while they use unreleased models which were trained in a similar way to GPT-3 (Brown et al., 2020).

We freeze the first 80% of the layers and the embedding and unembedding layers, as more recent work (Glaese et al., 2022; Menick et al., 2022) has shown that good results can still be achieved, and training is much more computationally efficient. In Table 14 we show the drop in performance for smaller OPT models between freezing 80% of the layers and not freezing. There is a drop of performance, but it is not catastrophic (equivalent to about a drop in model size among the three model sizes used), which justifies the tradeoff of training models with partially frozen weights.

## G    BEST OF N TEMPERATURE EXPERIMENT

In the main paper we report results for BoN using temperature 0.7. Here we show results for BoN with temperature 1, and show that temperature 0.7 performs better, hence our choice of it as the hyperparamter we use. Fig. 8 shows the results of BoN with two temperatures, as well as RLHF and SFT for comparison.

## H    SEQUENTIAL INSTRUCTIONS DATASET DETAILS

For the sequential instructions dataset, we build on the AlpacaFarm variant of the Self-Instruct protocol (Dubois et al., 2023; Wang et al., 2023), but adjust the seed instructions and prompt to gather more sequential instructions. Fig. 9 shows the prompt used to generate these instructions, and Table 5 shows examples of generated from the dataset.

```
You are asked to come up with a set of 20 diverse task instructions.
These task instructions will be given to a GPT model and we will
evaluate the GPT model for completing the instructions.

Here are the requirements:
1. Try not to repeat the verbs for each instruction to maximize diversity.
2. The language used for the instruction also should be diverse. For
example, you should combine questions with imperative instrucitons.
3. The type of instructions should be diverse. The list should include
diverse types of tasks like open-ended generation, classification, editing,
etc.
4. A GPT language model should be able to complete the instruction. For
example, do not ask the assistant to create any visual or audio output.
For another example, do not ask the assistant to wake you up at 5pm or
set a reminder because it cannot perform any action.
5. The instructions should be in English.
6. The instructions should be a sequential or compositional instruction
containing multiple steps, where each step is related to the previous
steps. Either an imperative sentence or a question is permitted.
7. Try not to repeat the verbs used for each part of the instruction
across instructions to maximize diversity.
8. The output should be an appropriate response to the instruction and
the input. Make sure the output is less than 100 words.

List of 20 tasks:
```

Figure 9: The prompt for text-davinci-003 to produce instructions for the sequential instructions dataset using the Self-Instruct protocol (Wang et al., 2023).

Table 5: Example inputs from the sequential instructions dataset

| |
| --- |
| Generate a list of items that may be found in a first aid kit, along with description on why each item is important. |
| Sort a list of emotions (sadness, joy, anger, fear, disgust) into two categories, and explain why each emotion fits into the categories created. |
| Explain the concept of Renormalization Group in simple words, describe three uses for Renormalization Group in theoretical physics, and discuss its relationship with scaling laws. |
| Summarize the history of the Cold War, explain the outcome of the war, and discuss its significance to the world today. |

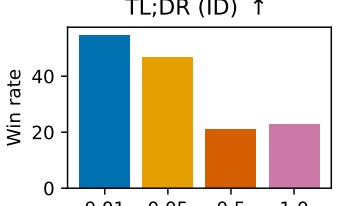 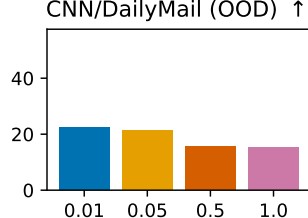 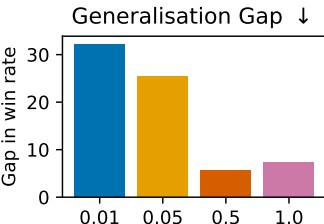

Figure 10: GPT-4 API evaluation win rate vs reference (text-davinci-003) outputs for RLHF models, based on LLaMa 7B, trained on the summarisation task, sweeping over KL penalty coefficient. In-distribution is performance on TL;DR, out-of-distribution is on CNN/DailyMail, and generalisation gap is ID – OOD performance.

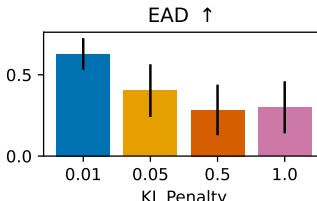 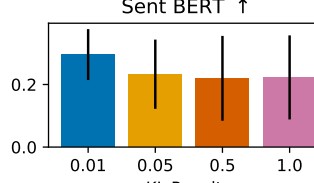 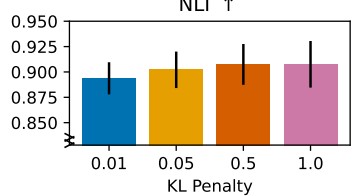

Figure 11: Per-input diversity metrics for RLHF summarisation models with different KL penalty coefficients. For these scores the outputs used to calculate the diversity are a sample of outputs from the model for single input. These per-input scores are then averaged, as in Eq. (2). Error bars are standard deviation of the per-input diversity score across different inputs. Note that some plots have broken y-axis for better visualisation.

# I  SUMMARISATION KL SWEEP RESULTS

Here we present results for generalisation and diversity for LLaMa models trained with RLHF on the summarisation task, sweeping over the KL penalty coefficient. This hyperparameter determines the weight of the KL penalty in the RLHF reward (the $\beta_{KL}$ in Eq. (1)). It could be the case that this KL penalty can control the tradeoff between diversity and generalisation, and so we try multiple different values of the penalty and include the results here.

We found that higher KL penalties actually resulted in less output diversity (Figs. 11 and 12), and also generally worse performance (Fig. 10), showing that the choice of KL penalty did not seem to provide a way to tradeoff between diversity and generalisation.

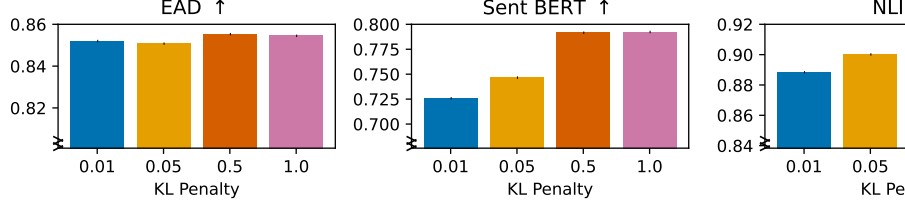

Figure 12: Across-input diversity metrics for RLHF, Bo16 and SFT policies. For these scores the outputs used to calculate the diversity are a set of single outputs from a range of inputs, as in Eq. (3). Note that all plots have broken y-axis for better visualisation.

Table 6: Size of the train, validation, test and OOD test datasets for each split for the SFT and RLHF models.

|  | length | sentiment | relationships |
|---|---|---|---|
| Train | 58770 | 58361 | 63324 |
| Validation | 3234 | 3223 | 3462 |
| Test | 3303 | 3276 | 3539 |
| OOD Test | 3250 | 3277 | 3014 |

## J  SUMMARISATION EXPERIMENTS WITH OPT

In addition to the experiments in the main paper, we did several experiments using OPT (Zhang et al., 2022) in the summarisation task with a different choice of ID and OOD test sets.

### J.1  DATASET SPLITTING

We create split versions of these datasets along several factors of variation in their inputs: *length*, *sentiment*, and *subreddit*. For each of these factors of variation, we create a train/test split where the train and test inputs are drawn from different parts of the distribution for that factor. For the *length* split, the training set consists of data where the post is less than 245 words (the median number of words in the SFT training distribution); for the *sentiment* split, we use an off-the-shelf sentiment classifier (Loria, 2013), and the training set consists of summaries with sentiment less than the median sentiment in the dataset; for the *subreddit* split, the training set consists of summaries from a specific subreddit, r/relationships. In all cases the test set is the complement of the training set in the full dataset, meaning that the trained models will be evaluated on inputs from a different distribution than the one seen during training.

In all cases, we apply the same splitting procedure to both the preference data and the input/output pairs, to ensure that the training and test sets are consistent across the different methods. Each of these splits was chosen to produce a roughly 50-50 split between the training and testing distributions. In the case of *length* and *sentiment* this is exact, and in the case of *subreddit* the r/relationships subreddit contains approximately 60% of the data.

While we do not expect these splits to capture the full range of distribution shifts models may experience when deployed, using a range of splits will give us a more robust measure of how well the policies trained with different methods generalise under distribution shifts.

The dataset we use from (Stiennon et al., 2022) (filtered from (Nallapati et al., 2016)) comes with train, validation and test splits, which we use throughout our work. For the SFT dataset these splits have size 116722, 6553 and 6447 respectively, and for the RM dataset they have size 92858, 33083, 50718 respectively. The SFT and RLHF splits are the same apart from the RLHF dataset does not require the summaries (outputs), just the posts (inputs). To create the dataset splits used for the OOD generalisation experiments in Section 6.1, we split each of the train, validation and test sets into an in-distribution (ID) and out-of-distribution (OOD) train, validation and test set. We then train on the ID train set, do model selection using the ID validation set, and evaluate on the ID and OOD test sets to measure the in-distribution and out-of-distribution performance.

For the sentiment split, we first measure the sentiment of each post using an off-the-shelf sentiment classifier (Loria, 2013). For a given subset of the dataset (i.e. train, validation, test), the ID version of that subset is the set of all inputs with posts whose sentiment is lower than the median sentiment, and the OOD version is the complement of that set. For the length split, we take the same approach using the length (in words) of the post, and the ID version of the subset is the set with posts of length less than the median length. For the relationships split, we take the ID version of the subset to be all posts in the r/relationships subreddit, and the OOD version to be the complement.

We apply this same splitting procedure to both the SFT and RM training datasets. Table 6 and Table 7 show the size of the training, validation, testing and OOD testing sets for the RLHF and SFT, and RM training, respectively. For the results in this work we randomly sample from the test and OOD test sets for those metrics, and we randomly sample from the validation set (which is in-distribution) for calculating metrics used for model selection.

Table 7: Size of the train, validation, test and OOD test datasets for each split for the reward models.

|  | length | sentiment | relationships |
|---|---|---|---|
| Train | 45395 | 46411 | 52346 |
| Validation | 16513 | 16529 | 17687 |
| Test | 25539 | 25353 | 27492 |
| OOD Test | 25180 | 25366 | 23227 |

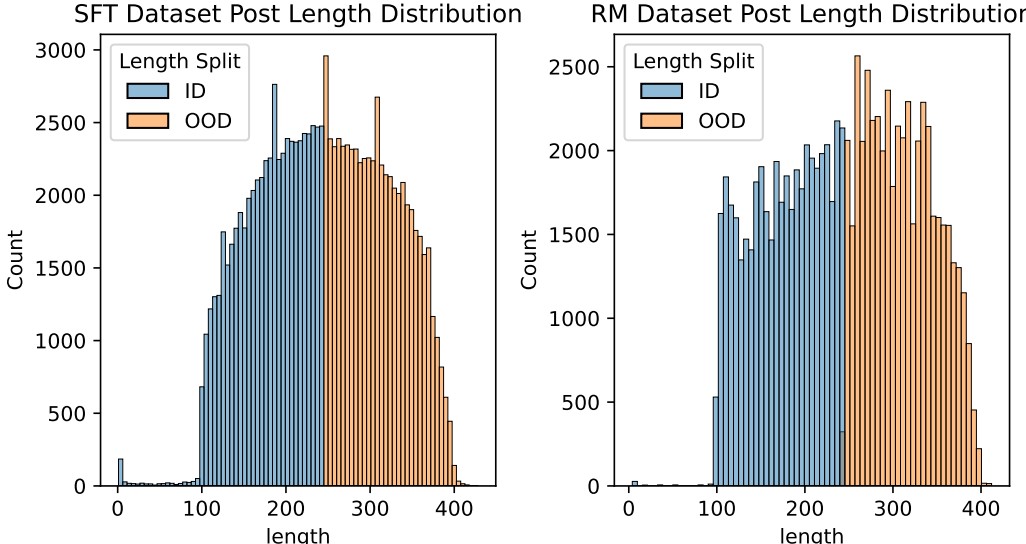

Figure 13: The distribution of post lengths across the full SFT and RM datasets. ID is the in-distribution version of the dataset, and OOD is the out-of-distribution version.

To understand the distribution shifts these splits entail, we show density plots for post length and sentiment across the full SFT and RM dataset in Fig. 13 and Fig. 14, and show the number of posts in each subreddit in Fig. 15.

## J.2 HYPERPARAMETERS

We use the same hyperparameters as in LLaMa training (see Appendix E.4), but sweep over learning rates for each model size. We detail the learning rates swept over for each model size and the chosen learning rate, for SFT, RM and RLHF training in Table 8, Table 9 and Table 10 respectively. In general for SFT and RLHF we did not see much variance with seeds, but we did in RM training, matching prior work (Stiennon et al., 2022). For the RLHF training with the largest two model sizes, due to the large amount of compute required to run multiple training runs, for several model size and dataset shift combinations we chose a single learning rate based on what we thought would give the best results at the time we started training. For the combinations where we did vary the learning rate we did not see much variation in performance on the metrics we measured, so we do not expect these choices to affect the results.

Table 8: Learning rates for different model sizes and dataset splits for SFT models. Underlined learning rate is the chosen one. $ke^{-n}$ means $k \times 10^{-n}$.

| Dataset | 125m | 350m | 1.3b | 2.7b | 6.7b |
|---|---|---|---|---|---|
| relationships | $\underline{1e^{-4}}$,$3e^{-5}$,$1.5e^{-5}$ | $\underline{1e^{-4}}$,$3e^{-5}$,$1.5e^{-5}$ | $\underline{1e^{-4}}$,$3e^{-5}$,$1.5e^{-5}$ | $\underline{1e^{-4}}$,$3e^{-5}$,$1.5e^{-5}$ | $1e^{-4}$,$3e^{-5}$,$1.5e^{-5}$ |
| length | $\underline{1e^{-4}}$,$3e^{-5}$,$1.5e^{-5}$ | $\underline{1e^{-4}}$,$3e^{-5}$,$1.5e^{-5}$ | $1e^{-4}$,$3e^{-5}$,$\underline{1.5e^{-5}}$ | $\underline{1e^{-4}}$,$3e^{-5}$,$1.5e^{-5}$ | $\underline{1e^{-4}}$,$3e^{-5}$,$1.5e^{-5}$ |
| sentiment | $\underline{1e^{-4}}$,$3e^{-5}$,$1.5e^{-5}$ | $1e^{-4}$,$\underline{3e^{-5}}$,$1.5e^{-5}$ | $1e^{-4}$,$3e^{-5}$,$\underline{1.5e^{-5}}$ | $\underline{1e^{-4}}$,$3e^{-5}$,$1.5e^{-5}$ | $\underline{1e^{-4}}$,$3e^{-5}$,$1.5e^{-5}$ |

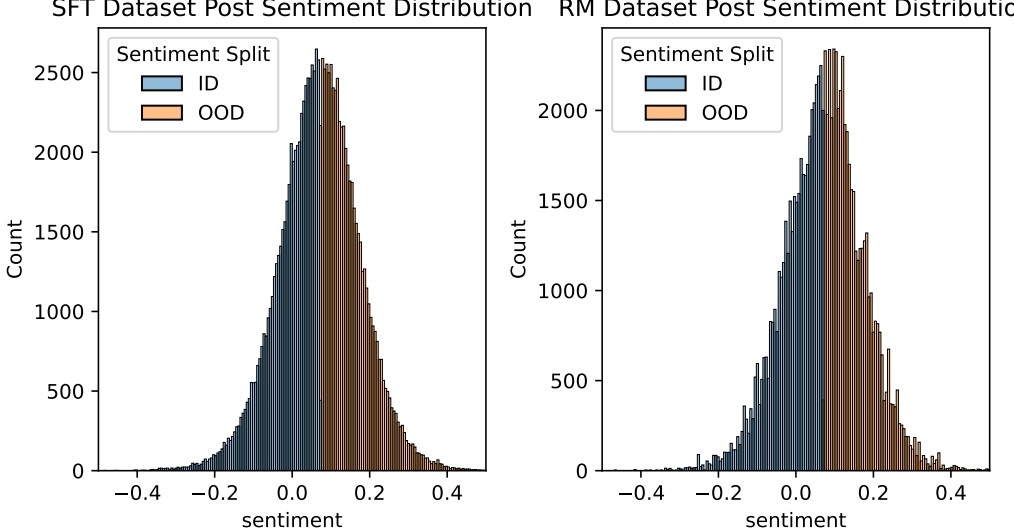

Figure 14: The distribution of post sentiment across the full SFT and RM datasets. ID is the in-distribution version of the dataset, and OOD is the out-of-distribution version.

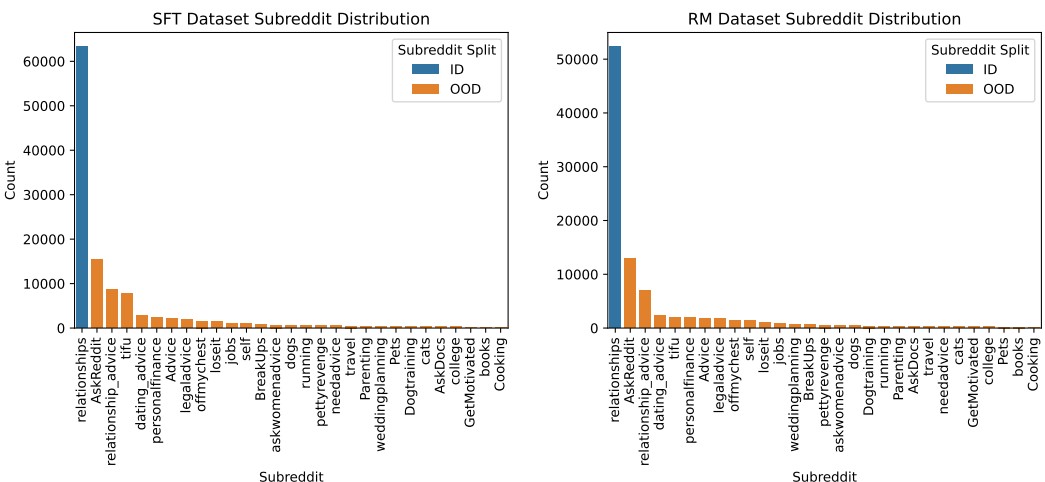

Figure 15: The number of posts in each subreddit across the full SFT and RM datasets. ID is the in-distribution version of the dataset, and OOD is the out-of-distribution version.

Table 9: Learning rates for different model sizes and dataset splits for reward models. Underlined learning rate is the chosen one. $ke^{-n}$ means $k \times 10^{-n}$.

| Dataset | 125m | 350m | 1.3b | 2.7b | 6.7b |
|---|---|---|---|---|---|
| relationships | $\underline{5e^{-4}},1.5e^{-5},5e^{-5}$ | $\underline{5e^{-4}},1.5e^{-5},5e^{-5}$ | $\underline{5e^{-5}},1.5e^{-5},5e^{-6}$ | $5e^{-5},\underline{1.5e^{-5}},5e^{-6}$ | $\underline{5e^{-5}},1.5e^{-5},5e^{-6}$ |
| length | $\underline{5e^{-4}},1.5e^{-5},5e^{-5}$ | $5e^{-4},1.5e^{-5},\underline{5e^{-5}}$ | $5e^{-5},\underline{1.5e^{-5}},5e^{-6}$ | $5e^{-5},\underline{1.5e^{-5}},5e^{-6}$ | $5e^{-5},1.5e^{-5},\underline{5e^{-6}}$ |
| sentiment | $5e^{-4},\underline{1.5e^{-5}},5e^{-5}$ | $5e^{-4},1.5e^{-5},\underline{5e^{-5}}$ | $\underline{5e^{-5}},1.5e^{-5},5e^{-6}$ | $5e^{-5},\underline{1.5e^{-5}},5e^{-6}$ | $\underline{5e^{-5}},1.5e^{-5},5e^{-6}$ |

Table 10: Learning rates for different model sizes and dataset splits for RLHF models. Underlined learning rate is the chosen one. $ke^{-n}$ means $k \times 10^{-n}$.

| Dataset | 125m | 350m | 1.3b | 2.7b | 6.7b |
|---|---|---|---|---|---|
| sentiment | $1e^{-4},3e^{-5},\underline{1e^{-5}}$ | $1e^{-4},3e^{-5},\underline{1e^{-5}}$ | $\underline{1e^{-5}},3e^{-6},1e^{-6}$ | $\underline{3e^{-5}}$ | $\underline{1e^{-5}}$ |
| length | $1e^{-4},3e^{-5},\underline{1e^{-5}}$ | $1e^{-4},3e^{-5},\underline{1e^{-5}}$ | $\underline{1e^{-5}},3e^{-6},1e^{-6}$ | $\underline{3e^{-5}}$ | $\underline{5e^{-6}}$ |
| relationships | $1e^{-4},3e^{-5},\underline{1e^{-5}}$ | $1e^{-4},3e^{-5},\underline{1e^{-5}}$ | $\underline{1e^{-5}},3e^{-6},5e^{-6}$ | $\underline{5e^{-6}},3e^{-6},1.7e^{-6}$ | $\underline{5e^{-6}},3e^{-6},1.7e^{-6}$ |

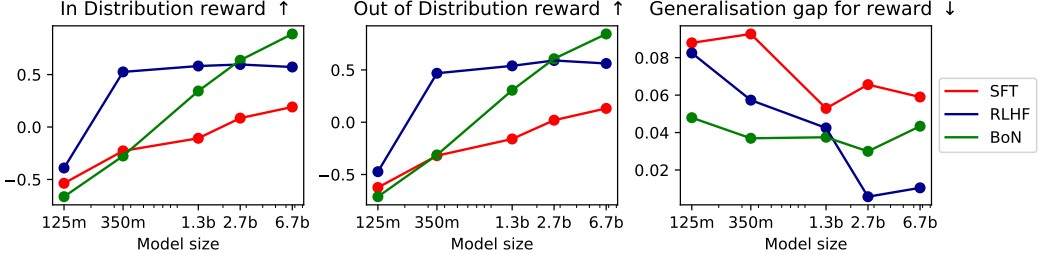

Figure 16: Proxy RM Score for SFT, BoN and RL models, averaged over dataset splits, for both in-distribution and out-of-distribution performance, and the generalisation gap. Arrows ↑, ↓ indicate whether higher or lower scores are better.

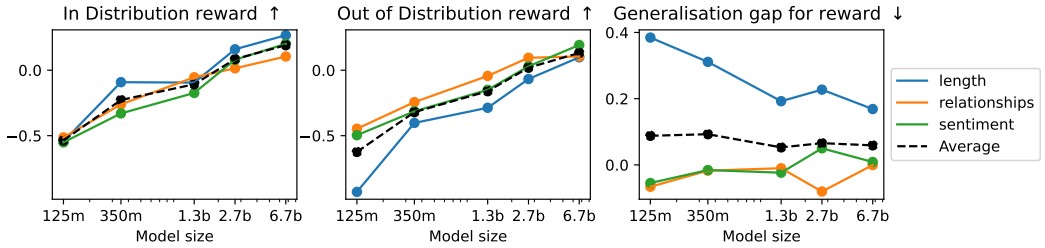

Figure 17: Proxy RM Score for SFT models for each dataset split, both in-distribution and out-of-distribution performance, and the generalisation gap. Arrows ↑, ↓ indicate whether higher or lower scores are better.

## J.3 GENERALISATION EVALUATION

For evaluation in these experiments, we train an RM as described in Appendix E.1 using the full dataset of summaries and preferences without splitting and the 6.7 billion parameter OPT model. We then use this *proxy RM* to evaluate the performance of SFT, BoN and RLHF models. As this reward model is trained with a different random seed and a different data distribution, it serves as a held-out automated evaluation of models, and a good proxy for human preferences.

We first discuss the results from the OPT version of the generalisation experiments described in Section 5.1. Fig. 16 shows the proxy RM score for SFT, BoN and RLHF models, averaged over dataset sizes. The important result here is that BoN generalises better than RLHF, which generalises better than SFT.

We see that at middling model sizes, RLHF outperforms BoN, but BoN scales better than RLHF, eventually outperforming it for models with more than 2.7b parameters. SFT comes out worst in this comparison, both scaling worse than BoN and the same as RLHF, and having worse absolute performance and generalisation. We see that BoN sampling does not see diminishing returns as model size increases, implying it will continue to be a useful yet simple technique. This experiment highlights the importance of training a reward model on human feedback and using it to select the best outputs at test time, potentially after fine-tuning the model.

**Supervised Fine-Tuning.** Fig. 17 shows the proxy RM score for the SFT models. Performance increases as model size increases, and in general performance drops OOD, which is unsurprising. There is a slight downward trend in the average generalisation gap across dataset splits as model size increases, implying that larger models fine-tuned with SFT generalise better (in terms of automated metrics). The relationships and sentiment splits produce negligible generalisation gap for the proxy RM score while the length split is more difficult.

**Best-of-N.** Fig. 18 shows the proxy reward score for the BoN models. Here we can see a smooth almost linear increase in performance as the model size increases. Given that the RM scores for smaller models were below chance, the fact the even for smaller models BoN results improve performance requires explanation. We hypothesise that the smooth increase here comes from two factors: the improvement in the SFT model being sampled from, and the improvement from the

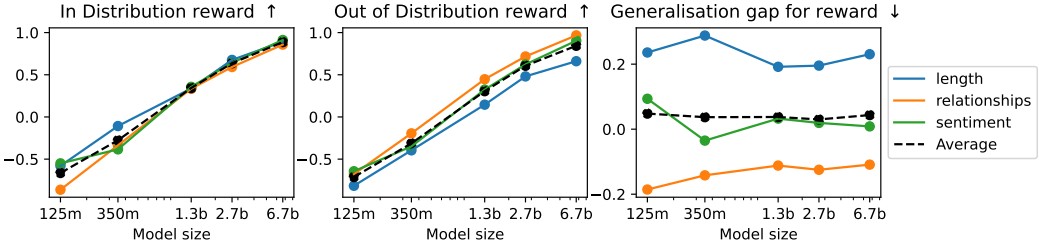

Figure 18: Proxy RM Score for BoN models for each dataset split, both in-distribution and out-of-distribution performance, and the generalisation gap. Arrows ↑, ↓ indicate whether higher or lower scores are better.

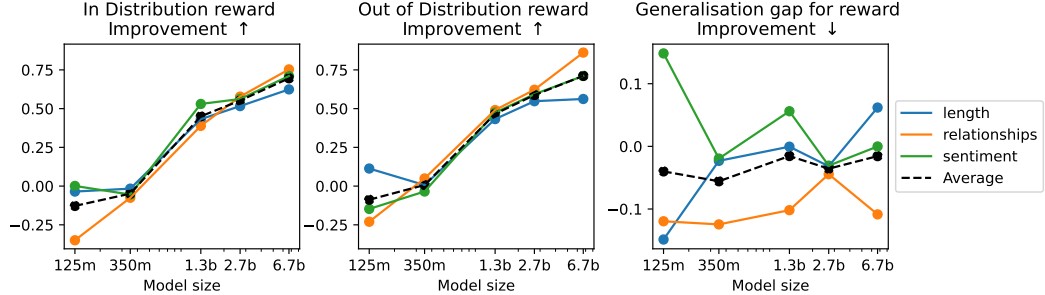

Figure 19: Proxy RM Score improvement using BoN on top of SFT models for each dataset split, both in-distribution and out-of-distribution performance, and the generalisation gap. Arrows ↑, ↓ indicate whether higher or lower scores are better. This plot is highlights the improvement from Fig. 17 to Fig. 18.

RM. At smaller model sizes increasing the number of parameters leads to improvements in SFT performance but not RM performance, while at larger model sizes increasing the number of parameters leads to improvements in both SFT and RM performance, but both to a lesser extent.

Fig. 19 shows the improvement of BoN over SFT. We can see that BoN only starts improving SFT as model size passes 350 million parameters, and the improvement grows as model size grows. This implies that as we increase model size BoN is likely to become a more performant choice compared to SFT. Further, we note that BoN uniformly improves the generalisation across model sizes (as shown by the black dashed line), implying that scaling models, even using SFT+BoN, will still result in a non-zero generalisation gap.

**Reinforcement Learning from Human Feedback.** Fig. 20 shows the proxy RM score for the RLHF models. Again, we see that increasing model size improves performance. Here we see a clearer trend of increasing model sizes reducing generalisation gap – this implies that as we make RL models larger, they are likely to generalise better. Given the difference between this trend and the generalisation gap trend for SFT models in Fig. 17, this partially justifies why RLHF is used in fine-tuning LLMs at a very large scale (Glaese et al., 2022; OpenAI, 2023; Ouyang et al., 2022): RLHF produces better-generalising models at larger model sizes than SFT.

## J.4 DIVERSITY EVALUATION

Table 11 shows the per-input diversity scores (Eq. (2)) for both RLHF and SFT models. The RLHF models have much lower diversity than SFT models according to all three metrics. While RLHF leads to better-generalising policies, those policies generate much less diverse outputs. We also see that diversity does not seem to change much with model size, apart from a slight downward trend for diversity in SFT models as model size increases.

Table 12 shows the across-input diversity scores (Eq. (3)). Here the corpus of text over which the diversity is measured is a single input sampled from the model for a range of outputs. Even if a model produces less diverse outputs for a single input, it could still produce different inputs for different outputs. This is the case for the EAD score, which is a proxy for diverse vocabulary and syntax, as

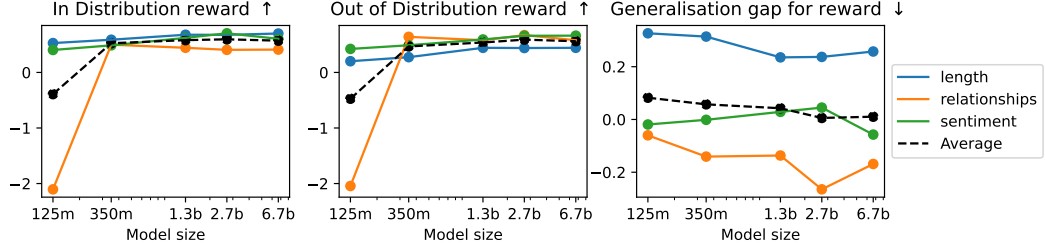

Figure 20: Proxy RM Score for RL models for each dataset split, both in-distribution and out-of-distribution performance, and the generalisation gap. Arrows ↑, ↓ indicate whether higher or lower scores are better.

Table 11: Per-input diversity scores for both RLHF and SFT models averaged over dataset splits. For these scores the outputs used to calculate the diversity are a sample of outputs from the model for single input. These per-input scores are then averaged, as in Eq. (2). Bolded results are better scores for each model size.

| Model Size | 125m | | 350m | | 1.3b | | 2.7b | | 6.7b | |
| Model Type | RLHF | SFT | RLHF | SFT | RLHF | SFT | RLHF | SFT | RLHF | SFT |
|---|---|---|---|---|---|---|---|---|---|---|
| EAD | 0.15 | 0.81 | 0.15 | 0.8 | 0.13 | 0.81 | 0.16 | 0.8 | 0.07 | 0.79 |
| Sent BERT | 0.15 | 0.5 | 0.12 | 0.46 | 0.15 | 0.48 | 0.13 | 0.45 | 0.06 | 0.45 |
| NLI | -1.08 | 0.26 | -0.96 | 0.2 | -1.0 | 0.12 | -1.07 | 0.09 | -1.54 | 0.06 |
| Average | -0.26 | **0.52** | -0.23 | **0.49** | -0.24 | **0.47** | -0.26 | **0.45** | -0.47 | **0.44** |

Table 12: Across-input diversity scores for both RLHF and SFT models averaged over dataset splits. For these scores the outputs used to calculate the diversity are a set of single outputs from a range of inputs, as in Eq. (3). Bolded results are better scores for each model size.

| Model Size | 125m | | | 350m | | | 1.3b | | | 2.7b | | | 6.7b | | |
| Model Type | RLHF | SFT | BoN | RLHF | SFT | BoN | RLHF | SFT | BoN | RLHF | SFT | BoN | RLHF | SFT | BoN |
|---|---|---|---|---|---|---|---|---|---|---|---|---|---|---|---|
| EAD | 0.69 | 0.86 | 0.85 | 0.87 | 0.87 | 0.86 | 0.86 | 0.87 | 0.86 | 0.86 | 0.87 | 0.86 | 0.87 | 0.87 | 0.86 |
| Sent BERT | 0.63 | 0.73 | 0.72 | 0.71 | 0.73 | 0.71 | 0.68 | 0.73 | 0.71 | 0.7 | 0.74 | 0.71 | 0.73 | 0.73 | 0.71 |
| NLI | 0.05 | 0.35 | 0.32 | 0.2 | 0.38 | 0.3 | 0.25 | 0.36 | 0.27 | 0.3 | 0.28 | 0.27 | 0.32 | 0.32 | 0.3 |
| Average | 0.46 | **0.64** | 0.63 | 0.59 | **0.66** | 0.62 | 0.6 | **0.65** | 0.61 | 0.62 | 0.63 | 0.62 | 0.64 | 0.64 | 0.62 |

Table 13: Across-input diversity scores for BoN models averaged over dataset splits. For these scores the outputs used to calculate the diversity are a set of single outputs from a range of inputs, as in Eq. (3). This is equivalent to Table 2 in the main paper but for BoN policies. See response to reviewer hdr6 for more details.

| Model Size | 125m | 350m | 1.3b | 2.7b | 6.7b |
|---|---|---|---|---|---|
| EAD | 0.85 | 0.86 | 0.86 | 0.86 | 0.86 |
| Sent BERT | 0.72 | 0.71 | 0.71 | 0.71 | 0.71 |
| NLI | 0.32 | 0.3 | 0.27 | 0.27 | 0.3 |
| Average | 0.63 | 0.62 | 0.61 | 0.62 | 0.62 |

Table 14: SFT Model ROUGE1 and reward model accuracy for the 3 smallest OPT model sizes, comparing freezing 80% of layers vs no layer freezing. Freezing generally results in less performance, as expected.

| Model Size | SFT Model Rouge1 | | RM Accuracy | |
| | 80% Frozen | 0% Frozen | 80% Frozen | 0% Frozen |
| --- | --- | --- | --- | --- |
| 125m | 0.217 | 0.221 | 0.482 | 0.496 |
| 350m | 0.2241 | 0.2233 | 0.498 | 0.508 |
| 1.3b | 0.221 | 0.2347 | 0.538 | 0.559 |

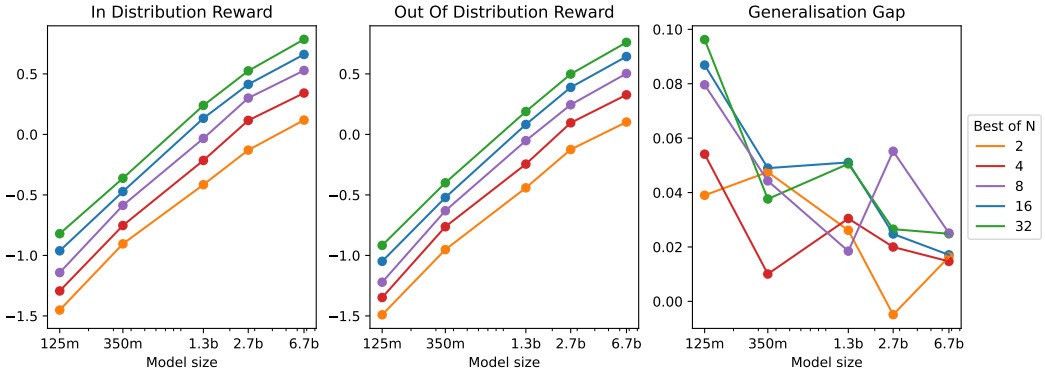

Figure 21: Proxy RM score for BoN sampling with varying $N$. All metrics are averaged over the 3 dataset splits.

for models with more than 125 million parameters both SFT and RLHF produce policies with very similar scores.

However, for the Sent BERT score, which is a proxy for diverse content and semantics, RLHF models are consistently less diverse than SFT models. This implies that RLHF produces models that have a tendency to generate outputs about certain topics or content regardless of the input. For the NLI score, which is a proxy for logical diversity, we see that as RLHF model size increases the score increases, eventually reaching the diversity of SFT models. Low NLI score for smaller models implies they have a tendency to make logically consistent claims in their outputs on top of producing outputs about certain topics of content.

### J.5 MODEL FREEZING EXPERIMENTS

We perform a small experiment to evaluate the effects of freezing the first 80% of layers during fine-tuning. The results are shown in Table 14, and show that while performance drops for the models evaluated in the experiment, the drop is not catastrophic, justifying the use of model freezing.

#### J.5.1 BoN PERFORMANCE FOR DIFFERENT N

In the previous OPT results we use $N = 64$ samples in best of N sampling. Here we show proxy RM scores for $N = 2, 4, 8, 16, 32$, to show how choices of $N$ trade off against performance. Fig. 21 shows proxy RM score in-distribution, out-of-distribution and the generalisation gap for these different choices of $N$. We see that increasing $N$ does lead to improved performance. At lower model sizes it leads to a larger generalisation gap, but this does not hold for larger model sizes.

## K   RLHF AND RM TRAINING CURVES

Here we present training curves for PPO and reward model training, for the summarisation task, for the models used in the main paper. In Fig. 22 we show the reward model validation accuracy throughout training. This is 1 epoch of training. In Fig. 23 we show the KL divergence and reward model score throughout training. PPO training has converged by approximately 250 PPO training steps, and so we terminated training early to save compute.

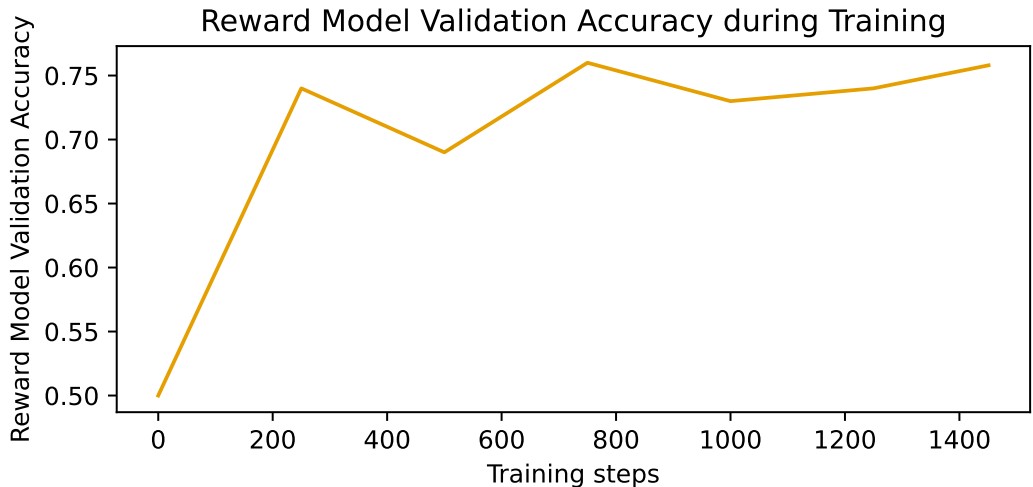

Figure 22: Reward model validation accuracy during training for the summarisation task.

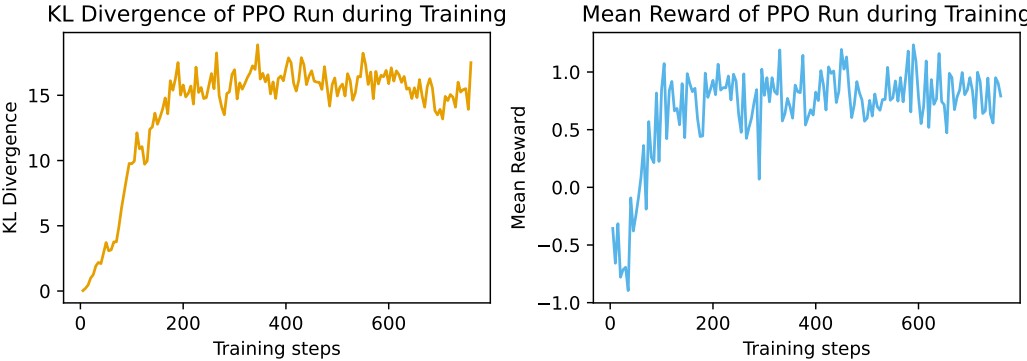

Figure 23: PPO KL divergence and reward model score curves for the summarisation task.

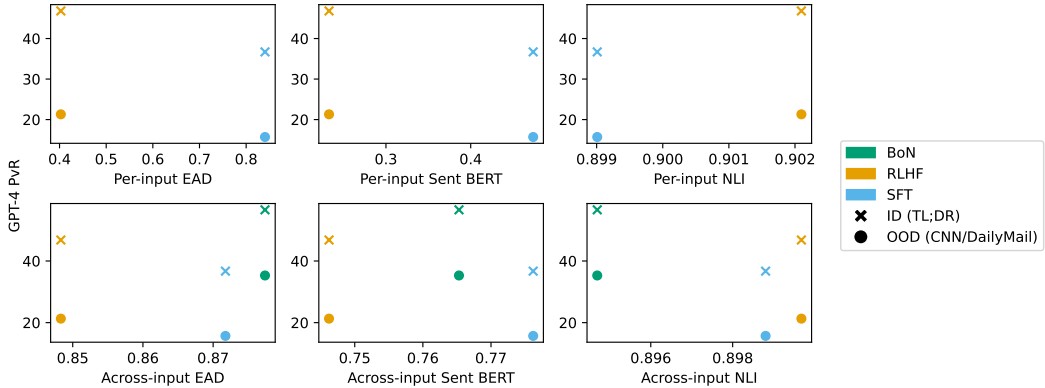

Figure 24: We plot the diversity vs gpt4 win rate trade-off for the summarisation task, across in-distribution and out-of-distribution winrates and per- and across-input diversity metrics.

## L    GENERALISATION VS DIVERSITY TRADE-OFF PLOTS

Fig. 24 shows the tradeoff between diversity and win rate in the summarisation task, across the three policy types we investigate. This reinforces the inherent tradeoff between generalisation and diversity present in existing language model fine-tuning techniques.

