# OpenReview forum: "Understanding the Effects of RLHF on LLM Generalisation and Diversity"
_ICLR.cc/2024/Conference — ICLR 2024 poster_

### Official Review · Reviewer_GaEc · 2023-10-22

**Soundness:** 2 fair
**Presentation:** 3 good
**Contribution:** 1 poor
**Rating:** 3
**Confidence:** 4

**Summary:**

This paper studies the effects of RLHF on generalization and diversity. Specifically, the authors look at the three stages in RLHF: supervised fine-tuning, reward modeling, and reinforcement learning. They conduct experiments that show that RLHF generalizes better than SFT to new inputs, but reduces output diversity.

**Strengths:**

The writing of the paper is clear and easy to follow. The paper studies three different aspects of performance, including in-distribution generalization, out-of-distribution generalization, and diversity. As far as I know, this covers a more comprehensive study on RLHF -ine-tuned model behavior than most observational studies in the literature.

**Weaknesses:**

This paper does not offer any new insight or novel methods compared to existing work in the literature, and no new methods have been proposed. First of all, the generalization capabilities offered by RLHF has been widely observed in state-of-the-art models, with clear comparisons and case studies of output from pretrained, instruction fine-tuned, and RLHF fine-tuned models (see, for example, the Llama-2 paper). The mode collapse phenomenon from RLHF has also been widely observed and measured. Maybe the only novelty this paper offers is evaluation on an array of sentence-level diversity metrics. Furthermore, the claims made in the paper are not very well-justified by experiment results, and some experiment details are lacking. Only two sets of experiments, namely summarization and instruction following, are conducted on one single model (Llama-7B), yet the paper makes a general claim about the effects of RLHF. More experiments on different models at potentially different scales could be helpful, but still, the contribution seems to be incremental.

My main concern is the contribution. Some additional questions are listed below for clarification, but unless the authors could justify their contribution through substantial experiments (on different models at different scales) and more in-depth analysis, I still lean towards rejection.

**Questions:**

The paper makes some unspecified claim that would need justification or further explanation. For example, on page 2, summary of contributions, the third bullet point: "...implying that such models tend to produce text of a specific style regardless of the input". How does one arrive at the "style" conclusion?

Why is there no error bars in Figure 5? Could you plot error bars over different choices of outputs from among the set of outputs to the same input?

Are the OOD datasets considered OOD for fine-tuning, or both fine-tuning and pretraining? The CNN/DailyMail dataset is most probably included in the pretraining dataset of Llama-7B.

---

> ### Author Response · Authors · 2023-11-16
> **Response Part 1**
>
> We thank the reviewer for their positive comments on the paper’s presentation and clarity. We’re glad you found the study comprehensive. We now address your comments in detail.
>
> > the claims made in the paper are not very well-justified by experiment results, and some experiment details are lacking. Only two sets of experiments, namely summarization and instruction following, are conducted on one single model (Llama-7B), yet the paper makes a general claim about the effects of RLHF. More experiments on different models at potentially different scales could be helpful, but still, the contribution seems to be incremental.
>
> In fact, **for the summarisation task we ran experiments with both LlaMA (7B) and OPT (125M, 350M, 1.3B, 2.7B, 6.7B) models**. The results with OPT are described in Appendix J. These results show the same trends as those with LlaMA, demonstrating that our conclusions generalise well across different base models, thus further strengthening our claims about the comparison between RLHF and SFT. While we cannot be entirely sure whether these conclusions hold in all possible settings, we conducted extensive experiments on **two different base models, two different tasks with five evaluation sets in total, across six different model sizes, using five different evaluation metrics for generalisation and diversity, resulting in training and evaluating over 80 models**. All these results point to similar conclusions. These experiments are very extensive, and required non-trivial resources. While it would always be better to have more experiments, we believe our analysis is rigorous and extensive enough to sufficiently support our conclusions.
>
> Following your suggestion, we will make sure to revise our claims to ensure we are not overclaiming and caveat that these conclusions are based on certain settings and models.
>
> In addition, we would like to note that the tasks we tested models on are some of the most commonly used in practical applications of LLMs, namely summarization and instruction following (which is also similar to single turn dialogue, another popular application), so we think our results can be of broad interest to researchers and practitioners alike. Even if our results only held in the instruction following setting this should still be considered valuable for the community. Instruction-following is clearly a very important and timely topic: it has a workshop at NeurIPS (https://an-instructive-workshop.github.io/), lots of interest both in academia and industry, and is one of the most common applications that LLMs are fine-tuned and then deployed for. Thus, we believe improving our understanding of methods even in this domain alone is useful and relevant for the ICLR community.
>
> [Response Continued Below]

---

> > ### Author Response · Authors · 2023-11-16
> > **Response Part 2**
> >
> > > the generalization capabilities offered by RLHF has been widely observed in state-of-the-art models, with clear comparisons and case studies of output from pretrained, instruction fine-tuned, and RLHF fine-tuned models (see, for example, the Llama-2 paper).
> >
> > While we believe our findings match intuition and anecdotal evidence about the characteristics of different fine-tuning methods for LLMs, we don’t believe our results have been shown in previous works. Could you please share the other works which show a similar result, so that we can include a comparison to them in our work and justify how our contributions are novel with respect to them? Note that In the “Generalisation and Diversity in NLP” paragraph in section 2 of the paper we discuss the differences with prior work both in terms of generalisation and diversity.
> >
> > For the paper you did mention, we will address how our results are different. In terms of generalisation, could you be specific about the parts of the LLaMa 2 paper that you think show that RLHF generalises better than SFT? There are results showing the RLHF improves on SFT training on test sets that are effectively in-distribution, and results showing the performance of RLHF on various safety benchmarks, but neither of these provide evidence comparing RLHF and SFT’s behaviour OOD. Our results involve creating specific OOD test sets (with respect to the fine-tuning dataset) which compare how models perform in situations they haven’t seen during fine-tuning. **We created an entirely new test dataset, sequential instructions, for this purpose, and show significant drops in performance as the test distribution becomes more OOD utilising this test set, which hasn’t been shown before**. The LLaMa 2 paper is concerned with producing the best model possible, and doesn’t hold out specific test sets to measure OOD generalisation, and performs limited comparisons between SFT and RLHF models.
> >
> > Even if our results are not necessarily surprising for practitioners frequently working with these models and methods, we believe it is important to demonstrate this empirically via thorough studies that others can later reproduce and build on. In addition, our studies emphasise the need for more research on methods that can strike a better balance between performance and diversity, thus opening up a new research avenue. To conclude, our core contributions consist of:
> > Demonstrating empirically that **RLHF produces policies that generalise better than SFT, and generalise better to harder distribution shifts, as shown by the generalisation performance on the novel Sequential Instructions dataset we introduced.**
> > Demonstrating that **this generalisation improvement comes at the cost of a large decrease in per-input output diversity, and smaller drop in across-input output diversity.**
> >
> > > The mode collapse phenomenon from RLHF has also been widely observed and measured.
> >
> > We are not aware of any papers demonstrating that the mode collapse phenomenon should be attributed to RLHF rather than SFT. Could the reviewer point us to such publications? In section 2 of the paper we discuss related work that shows some effects of RLHF on output diversity, but those works only look at per-input diversity, and use limited token-level metrics which typically don’t correlate well with human judgements of diversity [3]. In fact the original blog post highlighting the issue of mode collapse [1] was in fact not on an RLHF-ed model but rather an instruction-tuned model (text-davinci-002), so the question of whether RLHF differentially causes mode collapse is still open. Follow-on work [2] in fact seemed to imply that RLHF possibly doesn’t differentially cause mode collapse for OpenAI models.
> >
> > The LLaMa 2 paper you mentioned does a small experiment on diversity (in Figure 21), but this experiment is not as thorough as ours (it uses only 20 prompts in total), and it only uses the token-level Self-BLEU metric, which is insufficient for measuring a more nuanced and human-aligned notion of diversity compared to the range of metrics we use. In contrast, we use 100 prompts, we measure both per-input and cross-input diversity, and use 3 different diversity metrics that are supported by existing literature [3], all pointing to the same conclusion. We believe this breadth of evaluation setups is important to ensure claims regarding output diversity are well supported.
> >
> > [Response Continued Below]

---

> > > ### Author Response · Authors · 2023-11-16
> > > **Reponse Part 3**
> > >
> > > > The paper makes some unspecified claim that would need justification or further explanation. For example, on page 2, summary of contributions, the third bullet point: "...implying that such models tend to produce text of a specific style regardless of the input". How does one arrive at the "style" conclusion?
> > >
> > > Thank you for pointing out this lack of clarity, we agree it can be made more specific and justified. We have adjusted that bullet point to instead read:
> > > “Even when sampling outputs for different inputs, RLHF produces less diverse text on some metrics, implying that such models tend to produce more similar text regardless of the input.”
> > >
> > > To further explain what we intended to convey by this point: The reduction in across-input output diversity from RLHF compared to SFT implies that models’ distributions across inputs are less diverse. Taken to the limit, this would mean models would produce very similar outputs regardless of the input, and RLHF pushes models more towards this limit. The word “style” is just meant to mean outputs that are more similar.
> > >
> > > If there are other unspecified claims that you feel need more justification or explanation, we’d be happy to address those through adjustments to the paper.
> > >
> > > > Why is there no error bars in Figure 5? Could you plot error bars over different choices of outputs from among the set of outputs to the same input?
> > >
> > > We haven’t calculated those error bar statistics, but that is a good suggestion, thank you. We will run the experiments to calculate those error statistics and update the paper once those have completed, or add them to the camera ready if they don’t complete by the end of the author-reviewer rebuttal period. Unfortunately we will only be able to run them for the SFT and RLHF results, as for BoN each sample from the set of outputs costs 16x due to how BoN works (This is the same reason we don’t have BoN results for the per-input diversity), but this should still improve the results, especially as the main comparison is between RLHF and SFT.
> > >
> > > > Are the OOD datasets considered OOD for fine-tuning, or both fine-tuning and pretraining? The CNN/DailyMail dataset is most probably included in the pretraining dataset of Llama-7B.
> > >
> > > The OOD test datasets are considered OOD for fine-tuning, rather than pretraining. We believe this is still a relevant setting, as we often don’t have control or visibility of the pretraining data distribution. This is the “finetune-train/test” shift locus in [4], and in that survey they find 30% of work on generalisation in NLP is focused on this type of shift, making it a relevant and useful area for research.
> > >
> > > In conclusion, we hope we’ve addressed the concerns you have with the paper, particularly around novelty and contribution. We believe this work will be useful to the ICLR community and provides novel insights into the comparisons between these methods. Given this, would you be willing to increase your score? If you still believe the work to be lacking novelty, we’d especially appreciate specific pointers or references to where these results have been shown before.
> > >
> > > *[1] Mysteries of mode collapse, https://www.lesswrong.com/posts/t9svvNPNmFf5Qa3TA/mysteries-of-mode-collapse*
> > >
> > > *[2] RLHF does not appear to differentially cause mode-collapse, https://www.lesswrong.com/posts/pjesEx526ngE6dnmr/rlhf-does-not-appear-to-differentially-cause-mode-collapse*
> > >
> > > *[3] Evaluating the Evaluation of Diversity in Natural Language Generation, https://aclanthology.org/2021.eacl-main.25*
> > >
> > > *[4] State-of-the-art generalisation research in NLP: A taxonomy and review, https://arxiv.org/abs/2210.03050*

---

> > > > ### Author Response · Authors · 2023-11-20
> > > > **Revision with Error bars for Figure 5.**
> > > >
> > > > We mentioned in a previous comment we would update the paper in response to your question "Why is there no error bars in Figure 5? Could you plot error bars over different choices of outputs from among the set of outputs to the same input?"
> > > >
> > > > We have just uploaded an updated revision with Figure 5 updated with error bars. Figure 5 shows the across-input diversity, and for SFT and RLHF we calculate error bars using the standard deviation of the metric across different samples of the output for each input, as you suggested. Thank you again for the suggestion for improving the paper. The error bars are very small and non-overlapping, showing the difference in across-input output diversity is still present beween RLHF and SFT in this setting.

---

> ### Author Response · Authors · 2023-11-20
>
> Dear reviewer,
>
> We appreciate the time you have dedicated to reviewing our paper. In the response above, our responses have addressed your concerns regarding the novelty and clarity of our work, and so we hope you will consider raising your score. Otherwise, please let us know if there is anything preventing you from recommending acceptance of the paper.

---

> > ### Comment · Reviewer_GaEc · 2023-11-22
> >
> > Thank you for your detailed response. I apologize for missing the results on OPT models in the appendix. Most of my minor concerns are addressed. However, I am still concerned about the contribution of this work. It seems that the main insight of this paper is that RLHF helps the model generalize better at the sacrifice of output diversity. But this insight is not new. Many prior work have pointed out this issue and possible explanations [1-5].
> >
> > While I agree that a systematic study of the tradeoff between diversity and generalization of RLHF is interesting, I think this paper should not limit itself in confirming a phenomenon that has been observed widely and examined, if not explicitly, in ablation studies of many empirical works. At the minimum, could you plot the pareto frontier of diversity metrics versus OOD generalization across models and tasks? What would be potential ways that could push the pareto frontier outwards? Why is KL-divergence not sufficient in maintaining diversity? How about other entropy-based regularization?
> >
> > [1] Whose Opinions Do Language Models Reflect? Shibani Santurkar, Esin Durmus, Faisal Ladhak, Cinoo Lee, Percy Liang, Tatsunori Hashimoto. ICML 2023.
> >
> > [2] Aligning Language Models with Preferences through f-divergence Minimization. Dongyoung Go, Tomasz Korbak, Germán Kruszewski, Jos Rozen, Nahyeon Ryu, Marc Dymetman. ICML 2023.
> >
> > [3] Red Teaming Language Models with Language Models. Ethan Perez, Saffron Huang, Francis Song, Trevor Cai, Roman Ring, John Aslanides, Amelia Glaese, Nat McAleese, Geoffrey Irving. EMNLP, 2022.
> >
> > [4] A Distributional Approach to Controlled Text Generation. Muhammad Khalifa, Hady Elsahar, Marc Dymetman. ICLR 2021.
> >
> > [5] Improving alignment of dialogue agents via targeted human judgements. Glaese et al.

---

> ### Author Response · Authors · 2023-11-22
>
> Thanks for your response. We are glad to have addressed your minor concerns.
>
> We believe the insights provided by our paper are new, and especially that they have been shown robustly to hold across a variety of settings. To compare our experiments and conclusions to each of the works you shared:
>
> 1. We think the reviewer is pointing to the *Modal representativeness* paragraph of this work as evidence for a lack of diversity from RLHF. However, this experiment doesn't show that RLHF (as opposed to SFT, or some other part of the pipeline for training `text-davinci-003` that we don't know about) is the cause of this lack of diversity. Further, the experiment is limited to multiple choice question answering, and only considers one model.
>
> 2. This work does show that something similar to RLHF (RKL-DPG in that work) does reduce per-input output diversity and increases in-distribution reward. However, the metrics for diversity are limited, they don't investigate across-input output diversity, then don't compare RLHF to SFT or BoN, and they don't look at out-of-distribution generalisation. Further, their work is on much smaller models and simpler tasks than the ones we investigate.
>
> 3. This work again shows that RLHF increases in-distribution reward but decreases per-input output diversity, and does compare to SFT in this setting. However, it again only uses limited diversity metrics, doesn't evaluate cross-input output diversity, and doesn't investigate out-of-distribution generalisation. Further, the red-teaming setting investigated in that paper is plausible quite different from standard settings like instruction-following that we investigate, making it unclear whether their results would hold in these other settings.
>
> 4. This work shows that RLHF (Ziegler in that work) decreases per-input diversity somewhat while increasing in-distribution reward, but it's experiments are on small models (GPT-2), they only investigate per-input diversity, they use limited diversity metrics, and they don't investigate out-of-distribution performance.
>
> 5. This work compares SFT, BoN and RLHF, but does not have any experiments on out-of-distribution generalisation or output diversity, and so is quite unrelated to our work.
>
> Overall, *none of these works investigate out-of-distribution generalisation*, *all of them use more limited diversity metrics*, and *none of them evaluate cross-input output diversity*. There has been no study comparing the tradeoff between *out-of-distribution* generalisation and output diversity (both across-input and per-input), and hence our insights concerning the trade-off between these metrics of importance are novel.
>
> > At the minimum, could you plot the pareto frontier of diversity metrics versus OOD generalization across models and tasks?
>
> We have added a plot to this effect in the summarisation setting, with ID and OOD winrates vs per-input and across-input diversity measures, in appendix L (Figure 24).
>
> > What would be potential ways that could push the pareto frontier outwards? Why is KL-divergence not sufficient in maintaining diversity? How about other entropy-based regularization?
>
> While we agree these are all interesting research questions, and we are interested in further research investigating them, we believe our work as it currently stands is sufficient for acceptance to ICLR, and that answering those research questions is future work. Even if you believe the work is not worthy of acceptance, would you consider raising your score from a 3 to a 5, given our changes addressing your minor comments and our response above?

---

### Official Review · Reviewer_6G1j · 2023-10-28

**Soundness:** 2 fair
**Presentation:** 3 good
**Contribution:** 3 good
**Rating:** 6
**Confidence:** 4

**Summary:**

This paper aims to study the effects of RLHF for fine-tuning LLMs, focusing on out-of-distribution generalization and output diversity metrics. Through empirical experiments, this paper finds that RLHF can outperform SFT in terms of out-of-distribution generalization but at the cost of a decrease in output diversity. Such observations may help in better applying RLHF or SFT in specific applications.

**Strengths:**

- This paper conducted extensive experiments to elucidate why RLHF behaves differently from SFT. The experimental setup is sound, and the empirical results may inspire future progress in this direction.

- The paper is well-writen and easy to follow in general.

**Weaknesses:**

- Missing Related Work

In fact, there is a theoretical comparison of RLHF and SFT-style methods in the framework of imitation learning [1]. Indeed, LLMs are imitating human speech. In that framework, RLHF corresponds to adversarial imitation learning (AIL) methods, and SFT corresponds to behavioral cloning (BC). To the best knowledge of the reviewer, that theoretical study reveals that AIL (RLHF) methods can have better out-of-distribution generalization performance than BC (SFT) because AIL methods optimize their policy on out-of-distribution states (prompts) and rigorously prove this phenomenon. I believe this related work is insightful for studying the advantages of RLHF over SFT, and this paper should be mentioned in the related work.

[1] Xu, Tian, et al. "On the generalization of adversarial imitation learning and beyond." *arXiv preprint arXiv:2106.10424* (2021).


- Typos and Writing Suggestions

1. There are two minus symbols in Equation (1).
2. It seems unusual to draw a conclusion in Section 6.3 while presenting empirical evidence in Appendix I.

**Questions:**

The major concerns stem from the fact that the empirical evaluation heavily relies on the training quality of each algorithm, and the reviewer is uncertain about whether RLHF is trained to a high standard.

**Question 1**: Do empirical conclusions heavily depend on the training status of the reward model and PPO? The reviewer observed that this paper freezes some layers of LLMs when using RLHF, which may limit the representation capacity. Thus, the reviewer questions whether the training of RLHF is of good quality. Could this paper provide the evaluation accuracy of the reward model and training curves of PPO?

**Question 2**: Why not use entropy as a metric of diversity (although existing evaluation methods are acceptable)?

**Discussion**: This paper mentions that "Future work should investigate why RLHF reduces the output diversity so much," and the reviewer would like to point out some observations: the optimal policy by RL algorithms is deterministic (i.e., less diversity), if there is no tie in the reward value, there is no KL penalty, and the optimization is done perfectly. When there is a KL penalty, a recent paper shows that this corresponds to "soft Q-learning" [2]. In that case, the reward model is optimized perfectly. Although the algorithm in [2] is not applicable to the true RLHF setting where we only have out-of-distribution prompts and no preference labels, the viewpoint in [2] is insightful for in-distribution training.

[2] Rafailov, Rafael, et al. "Direct preference optimization: Your language model is secretly a reward model." *arXiv preprint arXiv:2305.18290* (2023).

---

> ### Author Response · Authors · 2023-11-16
> **Response Part 1**
>
> We thank the reviewer for their positive comments. We’re glad you found the work well-written and empirically sound. We’ll now respond to your comments in detail.
>
> > The major concerns stem from the fact that the empirical evaluation heavily relies on the training quality of each algorithm, and the reviewer is uncertain about whether RLHF is trained to a high standard
>
> **Given the strong performance of the RLHF and BoN policies relative to SFT when evaluated by humans and GPT-4**, and the high accuracy of the reward models (in fact slightly higher than reported in [2] - our RM gets .76 while the best model in [2] gets ~.74) **we believe this sufficiently demonstrates that our RLHF models are trained to a high standard**.
>
> Further, our conclusions rely on the relative ranking of RLHF, SFT and BoN and gaps in performance between methods. **Even if our RLHF implementation is not fully optimal, our conclusions would still be correct, as an improved RLHF performance doesn’t change the ranking of methods** in terms of generalisation as it already performs best in absolute terms on all evaluation sets.
>
> > Do empirical conclusions heavily depend on the training status of the reward model and PPO? The reviewer observed that this paper freezes some layers of LLMs when using RLHF, which may limit the representation capacity. Thus, the reviewer questions whether the training of RLHF is of good quality.
>
> We agree that empirical conclusions will depend on the performance of the reward model and PPO algorithm, as well as on the performance of the SFT optimisation.
>
> For the summarisation experiments, we believe our implementation is high quality for all these components, as our results mostly reproduce those in the literature: Our reward models gets .76 validation accuracy compared to ~.74 in [2]; our gpt-4 preference win rate for RLHF is .46 and for SFT is .36 while [1] gets ~.57 for RLHF and ~.41 for SFT (note there are slight differences in how GPT-4 is prompted). Slightly lower numbers should be expected given that we are freezing 80% of parameters. On freezing of layers, this happens for SFT, reward model training and PPO, so is unlikely to affect one method more than another. Thus, we’d expect the conclusions of the paper to hold even if all parameters were fine-tuned. Also, work such as [3] shows that impressive capabilities such as long-form dialogue and improved adversarial robustness can still be achieved when freezing a similar proportion of layers. Finally, in table 14 in the appendix we present some comparisons of frozen and unfrozen SFT and reward models based on OPT, and show only a small drop in performance when freezing 80% of the layers, the same as in the rest of our experiments.
>
> For the instruction-following experiments, we use the models released in AlpacaFarm [5]. These models get good performance in both GPT-4 and human evaluations in both their work and ours, and we believe that their implementation is of high quality.
>
> > Could this paper provide the evaluation accuracy of the reward model and training curves of PPO?
>
> Thank you for the suggestion to improve the thoroughness of the paper. **We have updated the appendix and added a section (Appendix K) with the training reward and KL divergence curves for the PPO model and evaluation accuracy of the reward model**, in the summarisation task. The reward model (with 80% parameters frozen) still achieves 75.8% validation accuracy, which is approaching the maximum given the levels of inter-annotator agreement reported in [2] being close to 25%. The PPO model converges at a KL divergence of approximately 15, and a validation reward of approximately 0.8.
>
> For the instruction-following task we used models released in AlpacaFarm ([5]), so we don’t have access to training reward curves or reward model accuracy.
>
> [Due to character limits, the response continues in a child comment.]
>
> *[1]  Direct Preference Optimization: Your Language Model is Secretly a Reward Model, https://arxiv.org/abs/2305.18290*
>
> *[2] Learning to summarize from human feedback, https://arxiv.org/abs/2009.01325*
>
> *[3] Improving alignment of dialogue agents via targeted human judgements, https://arxiv.org/abs/2209.14375*
>
> *[4] TextGAIL: Generative Adversarial Imitation Learning for Text Generation, https://arxiv.org/abs/2004.13796*
>
> *[5] AlpacaFarm: A Simulation Framework for Methods that Learn from Human Feedback, https://arxiv.org/abs/2305.14387*
>
> *[6] Evaluating the Evaluation of Diversity in Natural Language Generation, https://aclanthology.org/2021.eacl-main.25*

---

> > ### Author Response · Authors · 2023-11-16
> > **Response part 2**
> >
> > [Continued Response]
> >
> > > It seems unusual to draw a conclusion in Section 6.3 while presenting empirical evidence in Appendix I
> >
> > We chose to present evidence in the appendix due to space constraints, and the main result of that experiment was easy to convey concisely without a supporting figure. Unfortunately we won’t be able to fit the results from appendix I into the main paper due to space constraints without removing key information, but we hope the manuscript is still satisfactory to the reviewer.
> >
> > > Why not use entropy as a metric of diversity (although existing evaluation methods are acceptable)?
> >
> > Entropy is often used as an auxiliary loss in RL, often motivated by avoiding deterministic policies, but it hasn’t been used much as a measure of diversity, and we believe it is inadequate for several reasons. Firstly, it’s not been validated to align well with human notions of diversity, in comparison to the metrics we use in the paper (as shown in [6]). Secondly, it’s unclear how to interpret different values of policy entropy in terms of what that means for how diverse the policy outputs will be for a human.
> >
> > > This paper mentions that "Future work should investigate why RLHF reduces the output diversity so much," and the reviewer would like to point out some observations. [...]
> >
> > We thank the reviewer for their observations, we agree that the fact that RLHF without a KL penalty has a deterministic optimal solution, and that is likely part of the reason why RLHF reduces output diversity so much.
> >
> > > In fact, there is a theoretical comparison of RLHF and SFT-style methods in the framework of imitation learning [1]. [...] I believe this related work is insightful for studying the advantages of RLHF over SFT, and this paper should be mentioned in the related work.
> >
> > We thank the reviewer for the relevant reference. We have updated the paper (in the extended related works section of Appendix C) to include a reference to that work and added a discussion inspired by your comments.
> >
> > > There are two minus symbols in Equation (1).
> >
> > Thank you for pointing out this mistake, we have corrected it in the updated draft.
> >
> >
> > In conclusion, we hope we’ve addressed all your concerns about the paper, particularly in terms of the quality of the RLHF training and the additional related work. We hope this means you will raise your score, but if you have any outstanding concerns we will be happy to address them here, in order to aid your decision.
> >
> > *[1] Direct Preference Optimization: Your Language Model is Secretly a Reward Model, https://arxiv.org/abs/2305.18290*
> >
> > *[2] Learning to summarize from human feedback, https://arxiv.org/abs/2009.01325*
> >
> > *[3] Improving alignment of dialogue agents via targeted human judgements, https://arxiv.org/abs/2209.14375*
> >
> > *[4] TextGAIL: Generative Adversarial Imitation Learning for Text Generation, https://arxiv.org/abs/2004.13796*
> >
> > *[5] AlpacaFarm: A Simulation Framework for Methods that Learn from Human Feedback, https://arxiv.org/abs/2305.14387*
> >
> > *[6] Evaluating the Evaluation of Diversity in Natural Language Generation, https://aclanthology.org/2021.eacl-main.25*

---

> ### Author Response · Authors · 2023-11-20
>
> Dear reviewer,
>
> We appreciate the time you have dedicated to reviewing our paper. In the response above, our responses have addressed your concerns regarding related work and the quality of the training algorithms, and so we hope you will consider raising your score. Otherwise, please let us know if there is anything preventing from more strongly recommending acceptance of the paper.

---

> ### Comment · Reviewer_6G1j · 2023-11-21
>
> Thanks for the detailed response.
>
> - The concerns about the quality of reward models are well-addressed.
>
> - I find the KL divergence to be quite large. According to the reviewer's experience, the LLM tends to overfit when the KL divergence is larger than 1, as observed while working with Llama2-7B on datasets such as full-hh-rlhf. I have reviewed the literature and found that the provided KL results are consistent with those reported in [1]. The provided figures seem to suggest that KL regularization does not prevent the achievement of high reward scores. Therefore, the reviewer remains skeptical about the PPO training, but can accept the current results.
>
> - A typo exists in Equation (4) in Appendix.
>
>  [1] Learning to summarize from human feedback, https://arxiv.org/abs/2009.01325

---

> > ### Author Response · Authors · 2023-11-21
> >
> > Thanks for your response. We're glad we've addressed the concerns about reward model quality. Thank you for pointing out the typo, we will fix that in an updated draft.
> >
> > For the KL divergence, as you point out, our KL divergence curves are consistent with those in the literature. Our PPO training reward curves also are similar to those in [1] (see Figure 11). KL regularisation is meant to prevent achievement of any high reward scores, but rather to prevent overoptimisation, which we believe the chosen KL penalty has done. The correct KL penalty and the KL-reward tradeoff is very likely to vary between tasks given the difficulty of the task and the quality of the preference dataset, so we believe drawing inferences from experience on full-hh-rlhf isn't that informative. Hence, we believe our KL divergence is reasonable, and our PPO training is high-quality. Could you be more specific about why you remain skeptical of the PPO training, given that our training and evaluation results are consistent with prior work in the same setting for summarisation? What evidence would persuade you that it was high-quality?
> >
> > Further, as we mentioned in the original comment, our conclusions are mostly based on comparisons between RLHF and SFT, and as RLHF outperforms SFT in absolute terms across both datasets in the summarisation setting (in terms of GPT-4 and human preference winrate), we believe even if our PPO implementation isn't perfect, our conclusions would still be valid. Do you think a higher-quality PPO implementation would change the conclusions of our work?
> >
> > Otherwise, if you are happy to accept the current results and conclusions of the paper as valid and sound, we ask you to raise your score, or to let us know if there is anything else preventing you from raising your score at this stage.
> >
> > [1] Learning to summarize from human feedback, https://arxiv.org/abs/2009.01325

---

> > > ### Comment · Reviewer_6G1j · 2023-11-22
> > >
> > > Thanks for the detailed explanation.
> > >
> > > To clarify, I am not suggesting that a high-quality implementation (or optimization) of baseline methods would change the conclusion about the OOD generalization. At the current stage, this optimization curve is puzzling to me. It seems that the KL penalty does not work well. This may be due to the training configuration or datasets. Note that I do not deny the value of the conducted empirical results. Instead, I value them and suggest that we should be cautious and skeptical when drawing conclusions from empirical observations, for example, as in Section 4.3.
> > >
> > > I will make the final decision after discussing with other reviewers.

---

> > > > ### Author Response · Authors · 2023-11-22
> > > >
> > > > Thanks for your comment. We're glad you value our empirical results.
> > > >
> > > > We would like to better-understand what is puzzling about the optimization curve, and in what you think the KL penalty does not work well.
> > > >
> > > > * We think the optimization curves for KL and reward during PPO training are fairly standard and what is seen in other works (this open-source reimplementation in a similar settings shows a similar reward curve: https://wandb.ai/carperai/summarize_RLHF/reports/Implementing-RLHF-Learning-to-Summarize-with-trlX--VmlldzozMzAwODM2, as does [1], [2] (figure 4), [3] and [4]; the last two are somewhat different settings where the trends still hold). Do you think these curves aren't standard? If so, what do you expect them to look like, and why? Could you point to work that shows how you expect the curves to look?
> > > >
> > > > * In these curves, the KL penalty has it's intented purpose of limiting the growth of the KL divergence, as the model eventually converges to a KL of around 15 rather than growing in an unbounded manner. For the results referenced in section 6.3 and describe in appendix I, the converged KL of the policy trained with 0.5 and 1 KL penalty coefficients was 7.5 for both, and for the policy with KL penalty coefficients 0.01 is 31, which is again what we would expect from previous work and how the KL penalty is designed to work. What do you mean when you say the KL penalty does not work well? What behaviour would you expect to see, and why?
> > > >
> > > > > Note that I do not deny the value of the conducted empirical results. Instead, I value them and suggest that we should be cautious and skeptical when drawing conclusions from empirical observations, for example, as in Section 4.3.
> > > >
> > > > We assume you mean section 6.3 here. If so, we think the claims in that section are supported by the results in appendix I, and don't make unsupported conclusions. Are there specific parts of section 6.3 you think are too strong given the empirical evidence? We would be happy to work to adjust those claims so you think they are well supported by the experiments we have.
> > > >
> > > > ---
> > > >
> > > > [1] Learning to summarize from human feedback, https://arxiv.org/abs/2009.01325
> > > >
> > > > [2] Training a Helpful and Harmless Assistant with Reinforcement Learning from Human Feedback, https://arxiv.org/abs/2204.05862
> > > >
> > > > [3] Scaling Laws for reward model overoptimization, https://arxiv.org/abs/2210.10760
> > > >
> > > > [4] Reward Model Ensembles Help Mitigate Overoptimization , https://arxiv.org/abs/2310.02743

---

### Official Review · Reviewer_De3y · 2023-10-31

**Soundness:** 3 good
**Presentation:** 4 excellent
**Contribution:** 3 good
**Rating:** 8
**Confidence:** 4

**Summary:**

This paper empirically investigates the difference in generalization and generation diversity for LLMs trained with supervised learning and reinforcement learning for text summarization and instruction following. Moreover, they evaluate best of N (BoN), a very strong method for text summarization, as an additional method to test generalization of language models. They ultimately find evidence for RLHF improving generalization over supervised learning but at the cost of generation diversity.

**Strengths:**

- Their thorough investigation of RLHF vs SFT generation quality is very valuable. This work helps improve our understanding of why RLHF policies have empirically seemed to perform well in practice with users where more OOD data is likely encountered.
- The paper is very clearly presented and investigates two popular settings for RLHF finetuning.

**Weaknesses:**

Minor Comments:
For summarization, it appears that pretrained models already perform very well for CNN daily mail. Would the same diversity, generalization, performance relationships be seen when evaluating OOD performance on a different summarization dataset where Llama2 7B does not perform as well? Or perhaps more simply, when trained on CNN as in-distribution, how is OOD performance to the harder TL;DR task?

**Questions:**

Please refer the questions posed in the weaknesses section.

---

> ### Author Response · Authors · 2023-11-16
>
> We thank the reviewer for their positive review. We’re glad they found our investigation thorough and valuable, and the paper clearly presented. We now respond to your comments on the summarisation task.
>
> Firstly, for clarification and to avoid confusion, we’re using the LLaMa 7B model, not LLaMa 2 7B, although your comments are still valid. Secondly, are you aware of any papers that illustrate the performance of these pretrained models on CNN daily mail? We would expect them to do reasonably well, but not as well as models after fine-tuning. For example figure 4 in [1] shows pretrained models still doing comparably to transferred SFT models but much worse than RLHFed models in a similar setting (train on TL;DR, test on CNN/DM), although how the pretrained models were trained is not discussed.
>
> In your comment you imply that TL;DR summarisation is harder than CNN/DM summarisation. We haven’t seen evidence for this in our work or in previous works; could you share where you’ve seen this result so that we can cite and discuss it in our work? We might expect CNN/DM to be a harder summarisation task, as the inputs to summarise are much longer, and are likely to be more densely packed with information (as news articles) that is hard to summarise concisely as compared to reddit posts.
>
> Finally, it’s unfortunately not possible with our current resources to train RLHF models on CNN/DM since it would require collecting a lot of high-quality human preference data for training a reward model on this dataset which is expensive to do. The preference datasets we use were open-sourced by OpenAI in [1], but they didn’t release a large enough preference dataset for CNN/DM to enable reward model training. We agree it would be an interesting experiment to run, and we thank the reviewer for their suggestion. Our prediction is that the results would be the same: RLHF outperforms SFT both in-distribution and out-of-distribution, but at the cost of output diversity.
>
> We thank the reviewer again for their complementary review and useful comments. We hope we've answered all your concerns, but please let us know if you have any remaining questions.
>
> *[1] Learning to summarize from human feedback, https://arxiv.org/abs/2009.01325*

---

> ### Comment · Reviewer_De3y · 2023-11-20
>
> Thank you for the thorough response. I correct my statement that CNN is an easier summarization task. As the authors stated, when seeing prior work such as [1], we see that CNN/DailyMail summarization is a more difficult summarization task than TL;DR.
>
> I also understand and agree with the authors' point about the lack of high-quality preference data for CNN barring them from performing the same analysis that they did on the TL;DR summarization task.
>
> I will be maintaining my score.
>
> [1] Statistical Rejection Sampling Improves Preference Optimization, Liu et al 2023

---

### Author Response · Authors · 2023-11-16
**Global Response**

We thank all the reviewers for their informative and useful reviews. We have uploaded an updated draft of the paper with changes made to address concerns raised by the reviewers, with the changes coloured blue. We look forward to a productive and informative discussion period.

---

> ### Author Response · Authors · 2023-11-20
> **Revision with error bars for Figure 5**
>
> We have just uploaded an updated revision with Figure 5 updated with error bars. Figure 5 shows the across-input diversity, and for SFT and RLHF we calculate error bars using the standard deviation of the metric across different samples of the output for each input.

---

### Author Response · Authors · 2023-11-23
**Summary of Author-Reviewer Discussion and Disagreement**

We want to thank all the reviewers again for their work reviewing our paper, and for engaging in useful discussion to understand and address their criticisms. All reviewers found the paper thorough and comprehensive, and the presentation high-quality. Reviewer De3y thought the work was very valuable, and reviewer 6G1j thought the experimental setup was sound and the empirical results enable future progress in this area. We believe the paper has been improved by the suggestions and comments of all the reviewers, for which we are grateful.

We would like to summarise the discussion and disagreements we still have with two of the reviewers. Reviewer 6G1j was most concerned with the empirical evaluation relying heavily on the quality of each training algorithm. After discussion, they remain concerned about the quality of our PPO implementation. Firstly, we note that these concerns are about the summarisation experiments, as the instruction following models are taken from AlpacaFarm [1]. Secondly, our PPO training reward curves and KL divergence curves are consistent with those in the literature, as is our preference win rate [2]. Finally, even if the RLHF implementation was not entirely optimal, our conclusions would still hold: the conclusions rely on the comparisons between RLHF and SFT, and RLHF always outperforms SFT, so improving RLHF performance more would not change the conclusions. We have presented these arguments and evidence to the reviewer, but we are still uncertain as to why the reviewer believes the PPO implementation is not high-quality, given all our empirical results both in training and testing are in line with previous works (see comment here: https://openreview.net/forum?id=PXD3FAVHJT&noteId=OTbqZzUCpE).

Reviewer GaEc’s concerns were centred on the novelty of the work. They claim that our work does not offer any new insights on top of existing work in the literature. After discussion, they cited several works which they claim shows the results we present in our paper. However, we disagree that any of these works show the full set of results and conclusions we have in our work (see comment here: https://openreview.net/forum?id=PXD3FAVHJT&noteId=1QKXrVeAuM). Specifically, *no previous work investigates out-of-distribution generalisation*; *no previous work investigates across-input output diversity*; and *all previous works uses only limited token-level diversity metrics* that have not been shown to align well with human notions of diversity, as our metrics have [3]. We have addressed all of the other concerns in their review (e.g. adding error bars to figure 5; pointing out our results on OPT models; adding a pareto plot of diversity vs generalisation in the summarisation task), along with addressing the novelty concerns. We believe our work is sufficiently novel to warrant acceptance to ICLR, and don’t agree with the reviewer that our results have been shown in previous works.

Overall, we believe the paper is worthy of acceptance, and disagree with the criticisms levelled by the two reviewers. Our work provides valuable and novel insights into the effects of different language model fine-tuning techniques, which is a very important and timely research topic, and we hope the area chair will see this and recommend acceptance of the paper so that our results can be shared with the ICLR community.

[1] AlpacaFarm: A Simulation Framework for Methods that Learn from Human Feedback, https://arxiv.org/abs/2305.14387

[2] Learning to summarize from human feedback, https://arxiv.org/abs/2009.01325

[3] Evaluating the Evaluation of Diversity in Natural Language Generation, https://aclanthology.org/2021.eacl-main.25

---

### Meta-Review · Area_Chair_wjQF · 2023-12-20

**Metareview:**

The paper presents an empirical study on effects of Reinforcement Learning from Human Feedback (RLHF) and Supervised Fine-Tuning (SFT) in Large Language Models, focusing on text summarization and instruction following. Reviewers appreciated the extensive analysis, clear presentation, and its contribution to understanding effects of RLHF and SFT. However, concerns were raised about the paper's lack of novelty, limited experimental scope (given the limited choices of models and tasks), and insufficient justification for its claims. While I do think these concerns are valid, I also believe the paper offers enough interesting empirical findings to be of use to the community and thus recommend acceptance.

**Justification For Why Not Higher Score:**

Insufficient justification for its claims, and limited range of models evaluated.

**Justification For Why Not Lower Score:**

Empirical understanding of SFT and RLHF can be of great value to the community.

---

### Decision · Program_Chairs · 2024-01-16

Accept (poster)